# Cavity-enhanced single artificial atoms in silicon

**Valeria Saggio** [1] ✉, **Carlos Errando-Herranz**[1,2], **Samuel Gyger**[1,3], **Christopher Panuski** [1], **Mihika Prabhu**[1], **Lorenzo De Santis**[1,4], **Ian Christen**[1], **Dalia Ornelas-Huerta**[1], **Hamza Raniwala**[1], **Connor Gerlach** [1], **Marco Colangelo** [1] & **Dirk Englund** [1]

Artificial atoms in solids are leading candidates for quantum networks, scalable quantum computing, and sensing, as they combine long-lived spins with mobile photonic qubits. Recently, silicon has emerged as a promising host material where artificial atoms with long spin coherence times and emission into the telecommunications band can be controllably fabricated. This field leverages the maturity of silicon photonics to embed artificial atoms into the world's most advanced microelectronics and photonics platform. However, a current bottleneck is the naturally weak emission rate of these atoms, which can be addressed by coupling to an optical cavity. Here, we demonstrate cavity-enhanced single artificial atoms in silicon (G-centers) at telecommunication wavelengths. Our results show enhancement of their zero phonon line intensities along with highly pure single-photon emission, while their lifetime remains statistically unchanged. We suggest the possibility of two different existing types of G-centers, shedding new light on the properties of silicon emitters.

Quantum emitters in silicon are promising candidates for scalable quantum information processing[1]. Unlike platforms based on different host materials, these systems excel in meeting multiple key requirements at once: (i) they emit directly into the telecommunications wavelength band, enabling long-distance communication transfer without the need for any frequency conversion, (ii) have shown great spin coherence properties[2,3], and (iii) remarkably, can leverage the maturity of the silicon microelectronics and photonics industry[4] to realize scalable devices. Two main approaches are commonly used to integrate quantum emitters into silicon. The first one uses single erbium dopants to create emitters with great spin and coherence features[5,6]. The enhancement of their photon emission rate has been demonstrated by integrating these emitters into silicon nanophotonic structures, thus realizing spin-photon coupling[7–9]. The second approach is based on the creation of a broad diversity of color centers in silicon substrates[10,11]. Initial reports have focused on the G-center[12,13], the T-center[2,14], and the W-center[15], and include the first optical

observation of an isolated spin in silicon[2], and the first isolation of single artificial atoms in silicon waveguides and their spectral programming[16]. These color centers are particularly interesting for their shorter lifetime (from a few to hundreds of ns)—and therefore higher brightness—compared to erbium-based emitters, which feature lifetimes of ~μs already under enhancement. Our work focuses on G-centers, which are theoretically anticipated to show high coherent photon emission when integrated into optical cavities with high quality factors $Q$ (~$10^6$) and small mode volume $V$ (<$0.1\lambda^3$, with $\lambda$ the wavelength in the material)[17]. In more detail, the modified local density of optical states in a cavity can increase the radiative emission fraction $\beta$ into a desired mode while suppressing emission into other modes:

$$\beta = \frac{(\alpha + F_P)\gamma_R}{(\alpha + F_P)\gamma_R + \gamma_0},$$ (1)

[1]Massachusetts Institute of Technology, Cambridge, MA, USA. [2]University of Münster, Münster, Germany. [3]KTH Royal Institute of Technology, Stockholm, Sweden. [4]QuTech, Delft University of Technology, Delft, The Netherlands. ✉e-mail: vsaggio@mit.edu

where $\gamma_R$ is the radiative rate of an emitter placed in a homogeneous medium with the same refractive index as the host material, $\gamma_0$ encompasses the rates for non-radiative transitions, and $0 < \alpha \le 1$ is a parameter depending on the specific device structure and geometry (see SI, Sec. 7). $F_P$ is the cavity Purcell factor, which in the case of perfect cavity-atom coupling is defined as[18,19]

$$F_P = \frac{3\lambda^3}{4\pi^2}\frac{Q}{V}. \tag{2}$$

As Eq. (1) shows, $\gamma_R$ is enhanced by a factor $(\alpha + F_P)$ when the emitter is placed in a resonant structure. This comes from considering both the emission into the cavity mode, as well as into all other non-guided modes (see SI, Sec. 7 for more details).

To enable efficient collection of the light emitted from the cavity, it is required that the cavity far-field emission is matched to the optical mode of interest—such as the mode of an optical fiber. This light collection is quantified using the coupling efficiency $\eta$. The net collection efficiency $\beta\eta$ defines the performance of a quantum network built with such devices. Accommodating both high $Q/V$ and high $\eta$ in a single device is a nontrivial design challenge that depends largely on the materials, the fabrication, and the operation wavelengths of the artificial atom of choice.

Here, we report on the integration of single artificial atoms into inverse-designed, $\eta$-optimized photonic crystal cavities. We show cavity enhancement of the zero phonon line (ZPL) intensity, while the excited state lifetime of our atoms remains substantially unmodified. Our results suggest the possibility of two different types of artificial atoms labeled as G-centers. During the preparation of this manuscript, we became aware of a work reporting cavity-coupled silicon color centers[20]. We will discuss the results and how this compares to our work throughout our manuscript.

## Results

Our device, illustrated in Fig. 1a, consists of single G-centers coupled to inverse-designed 2D photonic crystal cavities. The G-center is a quantum emitter formed by two substitutional carbon atoms and a silicon interstitial (Fig. 1b), and features a ZPL transition at 970 meV (1279 nm) in the telecommunications O-band along with an expected spin triplet metastable state[21,22], which has so far been observed in ensembles only (Fig. 1c, d).

Our cavities were designed following our previous work[23] to simultaneously achieve a target $Q/V$ while optimizing for vertical coupling $\eta$ by matching the emission to a narrow numerical aperture in the O-band. Figure 2 shows one of our cavity designs, including its near-field cavity mode (Fig. 2a) and its far-field scattering profile (Fig. 2b), with more than 70% of the emitted power simulated to radiate into an objective NA of 0.55. More information on the optimization can be found both in "Methods" and SI, Sec. 1. The fabrication of our device follows our previous work[16] with the addition of an underetch step, and is described in "Methods", Sec. A "Sample fabrication".

The measurements on our device were performed with a setup consisting of a home-built cryogenic confocal microscope featuring temperature and $CO_2$ gas control, and optimized for visible light excitation and infrared collection into a single-mode fiber (details in SI, Sec. 2). A scanning electron micrograph of a representative cavity in our chip is shown in Fig. 2c. Its reflectivity was characterized in cross-polarization[14,23,24] (details in SI, Sec. 3). Figure 2d shows a photoluminescence (PL) 2D scan of one of our systems, where the cavity-coupled artificial atom is evidenced by the infrared emission in the cavity center upon excitation with green light. The spectral signature of the PL, shown in Fig. 1d, features a ZPL centered at around 1279 nm and thus aligns with the previously reported G-center ZPLs[12,16,25].

For the cavities of interest, we measure quality factors of ~3700 and 2100, and center wavelengths around 1279 nm, near the G-center ZPL. The measured quality factors are much lower than $\mathcal{O}(10^6)$, which is what is expected from simulations and already measured similar samples[23]. We attribute this to the damage induced by the carbon implantation process. A more detailed discussion about this, as well as suggested approaches to improve or avoid this problem, can be found in "Methods", Sec. B "Cavity far-field optimization".

Figure 3 shows the experimental results confirming the presence of single artificial atoms in our photonic crystal cavities. The coupling between the atom and the cavity results in an enhancement of the atom's single-photon emission. In Fig. 3a, we observe linearly polarized PL emission from our system (see SI, Sec. 2 for more experimental details), indicative of coupling through the expected transverse electric cavity mode. Measuring the PL saturation under increasing excitation power yields that of a two-level emitter model (see Fig. 3b and SI, Sec. 4). We highlight here that the counts reported in the saturation curve are extracted from spectroscopy measurements. The corresponding intensity value measured with our SNSPDs at saturation is ~40 K counts/s. Although a direct comparison with other works on G-centers is not straightforward due to different experimental parameters, we can conclude that our emitter features a notably high single-photon count rate (see SI, Sec. 8). We further confirmed the addressing of a single artificial atom by demonstrating single-photon emission via a Hanbury–Brown–Twiss experiment. Our second-order autocorrelation results (Fig. 3c) show excellent antibunching with a

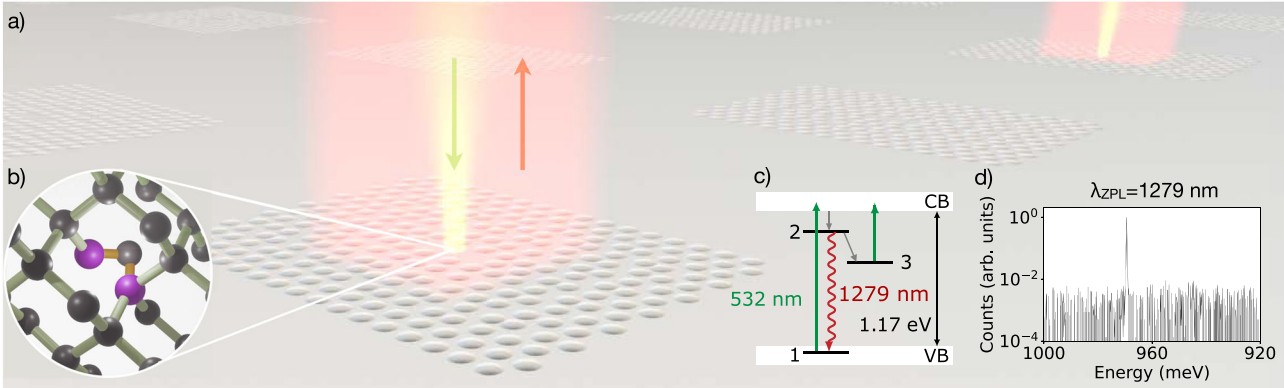

**Fig. 1 | Description of the system. a** Illustration of the system under study, consisting of optimized 2D photonic crystal cavities. **b** The G-center consists of two substitutional carbon atoms (purple spheres) and a silicon interstitial atom (gray spheres). **c** The O-band radiative transition occurs between singlet states 1 and 2 and can be excited via above-band excitation—from the valence band (VB) to the conduction band (CB). The system features an additional triplet state 3. **d** Spectrum showing the photoluminescence (PL) of a single cavity-coupled G-center with a ZPL around 1279 nm.

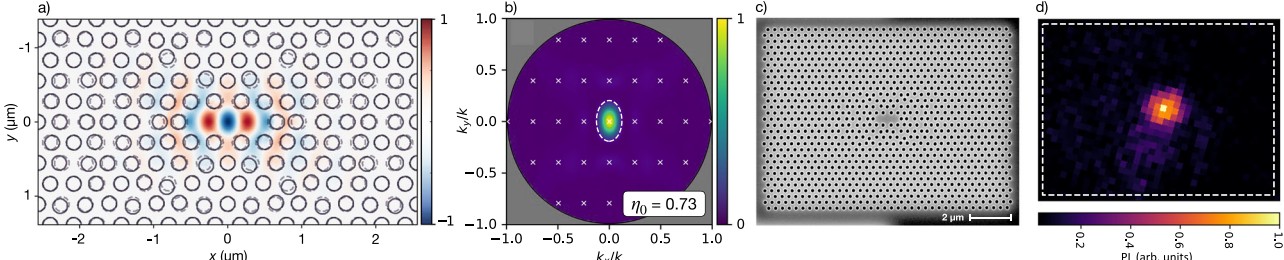

**Fig. 2 | Optimized cavities.** Simulated **a** near-field electric field amplitude and **b** scattered far-field power for the 2D photonic crystal cavity design used in this work, which maximizes vertical scattering as measured by the zero-order diffraction efficiency $\eta_0$ of tiled cavity unit cells (diffraction orders are marked by white × s). **c** Scanning electron micrograph showing one of our optimized photonic crystal cavities and **d** confocal PL 2D scan highlighting the PL emission from the cavity center.

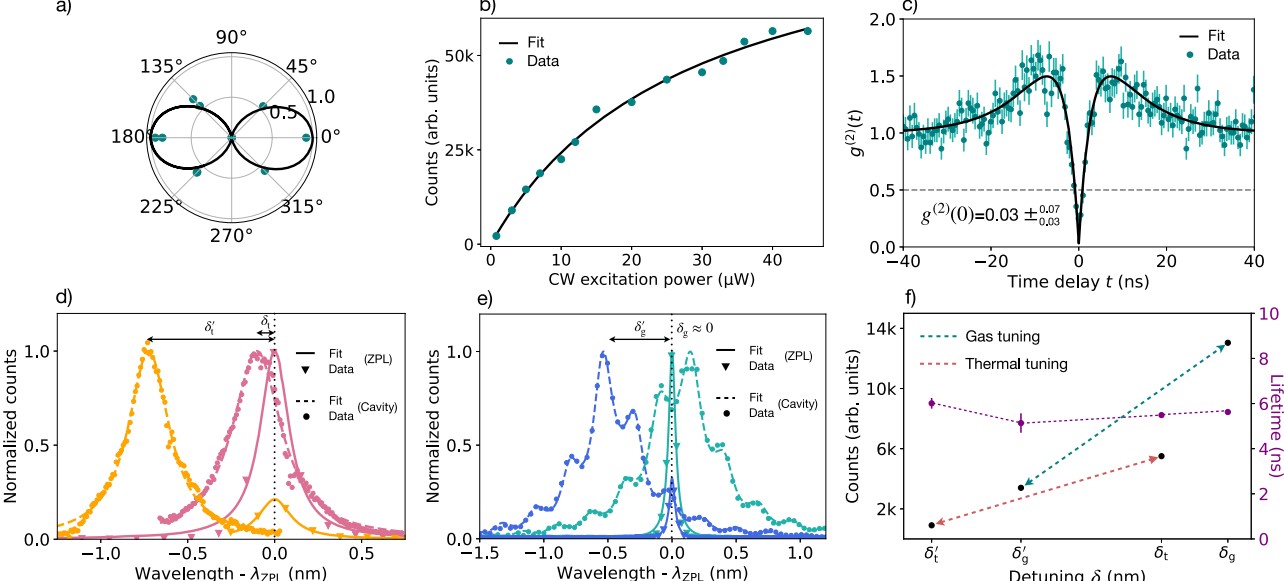

**Fig. 3 | Cavity-enhanced single-photon emission. a** Polarization plot of the PL emission from our system, which matches that of an electric dipole. **b** PL counts at increasing continuous wave (CW) excitation powers. Saturation matching that of a two-level system is observed. **c** The second-order autocorrelation function $g^{(2)}(t)$, which yields $g^{(2)}(0) \approx 0$, demonstrating high-purity single-photon emission. Poissonian error bars are associated with each data point. The error in the $g^{(2)}(0)$ is extracted from the fit. **d** Cavity-atom coupling is demonstrated by changing the sample temperature from 4 K (pink) to 24 K (orange) to spectrally tune a cavity with respect to the G-center ZPL, which results in a reduction of the PL magnitude. **e** This effect is confirmed with a second cavity-atom system on the same chip before (green) and after (blue) gas detuning. **f** PL counts and lifetimes for both systems (thermally and gas-tuned) under the measured cavity-atom detuning $\delta$ (nm). $\delta_t$ and $\delta_g$ indicate the detunings for the thermal and gas case, respectively, and the prime is used to differentiate between the two different detunings within the same case.

fitted $g^{(2)}(0)$ value of $0.03^{+0.07}_{-0.03}$ without background correction (this number refers to the gas tuning case and results lower than the $g^{(2)}(0)$ measured in the temperature tuning case thanks to a setup improvement, see details in SI, Sec. 5). This value is nearly an order of magnitude lower than the rest of the literature for G-centers (see SI Supplementary Table 1), and indicates high-purity single-photon emission. The bunching near ±10 ns delay conforms with the presence of a third dark state, which has been attributed to a metastable triplet state[21]. These measurements demonstrate the presence of a single G-center in our cavity.

To confirm the cavity enhancement of our single artificial atom, we detuned the cavity resonance wavelength from the G-center ZPL using two different methods, i.e., thermal and gas tuning. In the thermal tuning experiments, starting with a sample platform temperature of 4 K in our cryostat, we brought the temperature up to 24 K and consequently shifted the cavity away from our G-center ZPL. This effect is visible in Fig. 3d, where the pink (orange) curves show the cavity and ZPL profiles before (after) the temperature increase. Different detunings $\delta_t$ and $\delta_t'$, defined as the difference between the cavity and ZPL wavelengths, are therefore achieved at 4 K and 24 K,

respectively. While a significant cavity wavelength shift occurs, we also observe a much smaller ZPL shift, not shown in the figure (see SI, Sec. 6b for more details). Therefore, this plot shall be indicative only of the relative shift between the cavity and ZPL wavelength. A more detailed discussion about the figure can be found in SI, Sec. 6a. We note that temperatures below 30 K have been reported to not affect G-center ensembles[26,27]. We observe an intensity enhancement of $6.08 \pm 0.22$ with a cavity $Q \sim 3700$ and a mode volume of $V < 1(\lambda/n)^3$.

To validate that the intensity enhancement does not originate from the temperature change induced by thermal cavity tuning, we performed additional experiments using gas tuning. We injected $CO_2$ gas into the cryogenic sample chamber to coat the cavity with solid $CO_2$, followed by selective gas sublimation using a 532 nm CW laser. A further description of the process can be found in SI, Sec. 6b. Analogously to the thermal tuning case, Fig. 3e shows the cavity reflectivity and G-center ZPL now under gas tuning for two different detunings $\delta_g$ and $\delta_g'$. We calculate an enhancement in the PL emission of $3.84 \pm 0.07$ with a cavity $Q$ of $\sim 2100$ and a mode volume of $V < 1(\lambda/n)^3$. Also in this case, we observe a ZPL shift (not shown in the figure). We discuss how

this affects the tuning mechanism and provide additional data for several intermediate tuning steps in SI, Sec. 6b.

Ultimately, we measured the excited state lifetime of our artificial atoms under both gas and thermal tuning using a 0.5 ns pulsed laser at 532 nm for all cavity detunings (see SI, Sec. 4 for details about these measurements). Figure 3f shows our measured lifetimes and the emission rates for all of our experiments. The gas tuning data in Fig. 3f were acquired with a different spectrometer grating density compared to the data reported in Fig. 3b, hence showing lower counts compared to the saturation measurements (see SI, Sec. 4). We do not observe a statistically significant lifetime modification even under a clear enhancement of the zero-phonon emission rates above 6x.

The quantum efficiency (QE) of G-centers is a central question in the field. Recent reports have estimated the QE of a single G-center to be above 1% from waveguide-coupled counts[16,17], and <10% for ensembles coupled to separate cavities[28]. To gain a valid estimate of the QE of the G-center, an experiment varying the coupling rate between the same single artificial atom and a cavity was required. Our measurements allow us to extract such a value. Using the derivation described in SI, Sec. 7[28], the literature value of the Debye–Waller factor $F_{DW} = 0.15$[26], and our measured values of off-resonance lifetime $\tau_{off} = 6.09 \pm 0.25$ ns and count rate enhancement for thermal tuning, we obtain a QE bounded as QE < 18%.

## Discussion

We show intensity enhancement of G-centers' ZPLs coupled to silicon nanocavities, and highly pure and efficient single-photon emission. A central requirement for the scalability of our system is localized spatial and spectral alignment of both many cavities and many atoms to a common global frequency. The spatial alignment of the cavity and atom can be achieved by making use of localized implantation of single G- and W-centers[29]. Silicon artificial atoms can be spectrally aligned using the recently reported non-volatile optical tuning for G-centers[16] or methods used in other artificial atom systems such as tuning via electric fields[30], or mechanical strain[31]. Cavity tuning via local thermal oxidation of silicon has been achieved on a large scale[23], and a similar method could be used to align large arrays of cavity-atom systems at room or cryogenic temperatures. Our approach directly applies to other silicon artificial atoms, such as the T-center, which would enable direct access to a spin outside of a metastable state[2].

From Eq. (2), assuming $Q \sim 3000$ and $V \sim (\lambda/n)^3$, we can estimate a Purcell factor $F_P$ of a few hundred, which would be expected to lead to a perceivable lifetime reduction. In our experiments, however, no significant change in the excited state lifetime is observed. These results align with what was recently reported for ensembles of G-centers[28]. In general, the potential angular and spatial mismatch between the emitter's dipole moment and the cavity mode electric field would contribute to reducing the Purcell effect. However, we attribute the observance of constant lifetime mainly to the presence of strong non-radiative decay processes, which compete with radiative decay and thus effectively reduce the Purcell enhancement. Such non-radiative processes may be attributed to the presence of a metastable state in the G-center configuration, where the atom can decay without emitting photons. In principle, achieving higher quality factors (see "Methods", Sec. B "Cavity far-field optimization"), smaller mode volumes, and better cavity-emitter spatial alignment may induce enhancement that could outweigh non-radiative decay effects, thus resulting in lifetime reduction. This may also lead to a system with higher coherent photon emission, highly desirable for quantum information processing[17]. Lifetime shortening down to ~7 ns—comparable to our measured values even in the absence of lifetime reduction—was recently observed in ref. 20 for supposedly the same system. However, a hypothesis was recently raised regarding the possibility of two different physical systems being reported as G-centers[25]. Supplementary Table 1 in SI, Sec. 8 compares the reported

experimental results for single G-center labeled artificial atoms in silicon, and shows two clear clusters. The first group comprises the single emitter reports in refs. 12,16,17,25, and aligns with G-center ensemble work[28]. These studies show a ZPL centered around 1279 nm and a narrow inhomogeneous linewidth < 1.1 nm, a QE between 1% and 18%, and a short excited state lifetime < 10 ns that does not change significantly under Purcell enhancement, confirming the QE magnitude. The second group comprises refs. 13,20, and features a shorter ZPL centered around 1270 nm and a larger inhomogeneous linewidth of 9.1 nm, a QE up to unity, and a longer excited state lifetime > 30 ns, which changes significantly under Purcell enhancement and thus qualitatively aligns with the estimated QE. Our measurements align with the first system, i.e., the originally reported G-centers, and provide the first upper bound for the QE of single G-centers, previously estimated to be between 1%[16,17] and 10% for ensembles[28]. More information on this comparison can be found in SI, Sec. 8. While our findings hint at the potential existence of two distinct artificial atom systems, the observed variations could still stem from differences in measurement configurations, strain, or fabrication methods. We thus conclude that our work highlights the need for further theoretical and experimental investigation regarding the creation process and the photophysics of G-center-like artificial atoms in silicon platforms.

In summary, we showed cavity-enhanced single artificial atoms in silicon by integrating single G-centers into inverse-designed photonic crystal nanocavities. We demonstrated a 6-fold intensity enhancement, as well as the highest purity single-photon source for G-centers in the literature, and the first bound to the QE of single cavity-coupled G-centers of <18%. New directions may involve the investigation of the anticipated spin in the metastable state, which holds promise for various quantum applications such as quantum sensing and security protocol demonstrations. However, our device could already be suitable to implement spin-photon interfaces based on e.g., T-centers, which possess an addressable spin in the ground state[2]. Moreover, we shed light on new properties of single artificial atoms in silicon, suggesting the possibility of two different types of G-center-like structures observed so far. Despite G-centers—and artificial atoms in silicon in general—being still under investigation, the impressive amount of literature generated in the past few years suggests that emitters in silicon may indeed hold promise as viable candidates for practical integration and large-scale quantum information processing[1,32,33].

## Methods
### Sample fabrication

The fabrication process follows[13], starting from a commercial SOI wafer with 220 nm silicon on 2 μm silicon dioxide. Cleaved chips from this wafer were implanted with $^{12}C$ with a dose of $5 \times 10^{13}$ ions/cm$^2$ and 36 keV energy, and subsequently annealed at 1000 °C for 20 s to form G-centers in the silicon layer. The samples were then processed by a foundry (Applied Nanotools) for electron beam patterning and etching, resulting in through-etched silicon cavities with $SiO_2$ bottom cladding and air as top cladding. The silicon etching was performed using inductively coupled plasma reactive ion etching with $SF_6$–$C_4F_8$ mixed gas. As a final step, the samples were under-etched in a 49% solution of hydrofluoric acid for 2 minutes and dried using a critical point dryer.

### Cavity far-field optimization

Traditional photonic crystal cavity optimization aims to cancel radiative loss to enhance the quality factor $Q$, which also reduces the collection efficiency. To avoid this, we incorporate the far-field collection efficiency $\eta$ to the optimization objective function alongside maximizing $Q$ and minimizing mode volume $V$[23] (Fig. S3 in SI, Sec. 1). This process is implemented using the open-source package `Legume`[34] which maps the problem of cavity design onto efficient and auto-

differentiable guided mode expansion for gradient-based optimization.

As already mentioned in the main text, we observe quality factors much lower than the $Q \sim \mathcal{O}(10^6)$ expected from both simulation (Fig. S3 in SI, Sec. 1) and previous statistical studies on thousands of photonic crystal cavities designed for ~1550 nm operation under the same optimization method[23]. We attribute this disparity to the high carbon doping density used to produce cavity-coupled G-centers with sufficient probability. Reducing the doping density or applying localized doping[12] could play a role in recovering performance closer to intrinsic silicon. Applying large-scale characterization techniques[35] to locate ideal emitters and fabricate cavities around these positions could enhance the yield of coupled emitters in the case of reduced doping density.

## Reporting summary

Further information on research design is available in the Nature Portfolio Reporting Summary linked to this article.

## Data availability

All data sets generated in this study have been deposited in the Zenodo database under the accession code https://doi.org/10.5281/zenodo.11087252.

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

## Acknowledgements

The authors acknowledge Qiushi Gu, Kevin C. Chen, Chao Li, Camille Papon, Hugo Larocque, and Mohamed ElKabbash for advice and support. C.E.-H. and L.D.S. acknowledge funding from the European Union's Horizon 2020 research and innovation program under the Marie Sklodowska-Curie grant agreement no. 896401 and 840393. M.P. acknowledges funding from the National Science Foundation (NSF) Convergence Accelerator Program under grant no. OIA-2040695 and Harvard MURI under grant no. W911NF-15-1-0548. I.C. acknowledges funding from the National Defense Science and Engineering Graduate (NDSEG) Fellowship Program and NSF award DMR-1747426. M.C. acknowledges support from the MIT Claude E. Shannon Award. D.E. acknowledges support from the NSF RAISE TAQS program. This material is based on research sponsored by the Air Force Research Laboratory (AFRL), under agreement number FA8750-20-2-1007. The U.S. Government is authorized to reproduce and distribute reprints for Governmental purposes notwithstanding any copyright notation thereon. The views and conclusions contained herein are those of the authors and should not be interpreted as necessarily representing the official

policies or endorsements, either expressed or implied, of the Air Force Research Laboratory (AFRL), or the U.S. Government.

## Author contributions

V.S. contributed to designing, building, and running the experimental setup, carried out the experiment, and performed data analysis. C.E.-H. handled the sample fabrication, and contributed to the experimental setup, spectral data analysis, and theoretical interpretation of the results. S.G. contributed to setting up the experiment, and performing thermal-tuning measurements and lifetime data analysis. C.P. designed the photonic crystal cavities and sample. M.P., L.D.S., I.C., D.O.-H., H.R., and C.G. performed preliminary experimental characterizations. M.C. assisted with the single-photon detector measurement setup. C.E.-H. and D.E. conceived the idea. V.S., C.E.-H., C.P., I.C., and D.E. wrote the paper, with input from all authors.

## Competing interests

The authors declare no competing interests.
