## [Peer Review File · Nature Communications]

Cavity-enhanced single artificial atoms in siliconREVIEWER COMMENTS

Reviewer #1 (Remarks to the Author):

In the manuscript titled "Cavity-enhanced single artificial atoms in silicon," by V. Saggio et al., the authors present their research on cavity-enhanced single artificial atoms in silicon, which holds great potential for applications in quantum networks, scalable quantum computing, and sensing. The authors address the challenge of weak emission rates of these particular artificial atoms by coupling them to optical cavities and demonstrate the optimization of photonic crystal cavities for controllable cavity-coupling of single G-centers operating in the telecommunications O-band.

Silicon as a host material for single artificial atoms operating in the telecommunication bands has a unique potential to combine long spin coherence times with telecommunication wavelength photons, leveraging at the same time the success of silicon microelectronics and photonics. This makes the silicon platform of particular interest to the quantum photonics community.

The weak light-matter interaction leads to a low single-photon emission rate by quantum emitters in solids, which is typically below 1 GHz. Coupling of single-photon emitters to optical resonators allows to substantially enhance the light-matter interaction and speed up the emission rate. In this work, the authors address this challenge by coupling G-centers in silicon to 2D photonic crystal cavities. Notably, the authors apply the design approach to achieve high-quality cavities that offer both high Purcell enhancement, Q/V (quality factor per mode volume), and high coupling efficiency η . This not only enhances the single-photon generation but also ensures high collection efficiency, which both constitute the key performance metric of the single-photon source.

The paper provides experimental evidence for the successful coupling of single artificial atoms to the optimized cavities. The measurements confirm the presence of single G-centers in the photonic crystal cavities and demonstrate an enhancement of the emitter's single-photon emission.

Therefore, I judge this manuscript to be of interest to the Nature Communications readership and quantum information research community. However, before I can recommend publication, I have several questions (listed below) that must be addressed.

Comments:

1. How would the authors explain no change in the excited state lifetime? Clear lifetime shortening was reported in Redjem, W. et al. All-silicon quantum light source by embedding an atomic emissive center in a nanophotonic cavity. arXiv preprint arXiv:2301.06654 (2023)
2. Along those same lines, how do the saturation measurements of the emitter compare when it is in resonance vs when it is detuned from the cavity? In Figure 3f, the authors show an increase in counts on-resonance vs off-resonance, but these numbers are in arbitrary units, and much lower than the saturation measurement shown in Figure 3b. I would suggest that saturation be compared at different cavity detuning since the coupled emitter should have a high saturated intensity than when it is uncoupled.
3. What limits the quality factor of the fabricated photonic crystal cavities? The authors predict Q-factors of up to 5.0×10^5 theoretically but only measure $Q \sim 5000$.
4. How is the emitter dipole oriented relative to the cavity?
5. What is the fabrication yield of the devices? Based on the presented example as the best result, what potential approaches do the authors suggest for improving performance and achieving greater scalability that could be explored immediately?
6. The argument the authors make that this work is conclusive evidence of two closely related but different types of G-centers is somewhat lacking in evidence. For example, the authors claim that

Redjem et al show lifetime shortening of a different type of defect center since they report an uncoupled excited state lifetime of ~ 30 ns. However, Redjem et al report a ZPL wavelength of 1275 nm, which is close to the 1279 nm ZPL wavelength reported here; the discrepancy between the two could be explained by variations in strain originating from damage induced by implantation. The possibility of different color centers in Si due to carbon implantation should be investigated more systematically before such a claim can be made.

Nevertheless, this work is an important step in silicon-based quantum photonics using artificial atoms, and I judge it likely to be of broad interest. I recommend major revision as the best option.

Reviewer #2 (Remarks to the Author):

Saggio et al. report an experimental study on single-photon emitters that are embedded into a nanophotonic resonator made from silicon. The study is put into the context of distributed quantum information processing, or quantum networking, which is a topic that is of high relevance to a broad audience. Compared to other materials under study, such as quantum dots in GaAs or color centers in diamond or silicon carbide, the presented choice of emitter and host has several advantages: First, nanofabrication in silicon is a well-advanced technology, which may aid the up-scaling of corresponding devices. Second, emission into the telecom wavelength band allows for transmission over optical fibers with reduced loss. Albeit these improvements seem technical at first sight, they pose critical steps towards up-scaling of quantum technologies. Thus, the subject matter is in principle suited for publication in Nature Communications. However, the manuscript in the current form is not. The reason is that the achievements of the authors are heavily oversold, and the central claims of the paper are not backed by the measurements. Even worse, much of the information that is required to judge the achievements (or the shortcomings) of the manuscript is only accessible after studying the supplementary information in detail, while it does not show up in the main text.

In detail, the main criticisms are:

- 1) The paper is put into the context of quantum networking. The authors mention several times that this requires long-lived spins. Thus, one would expect that the experimental data includes some evidence for such long-lived spins. However, there is no corresponding measurement. Even after studying the supplement, I could not tell whether such spin is present or even expected in the G-center. Only after reading some of the references, I understood that the studied emitter does not possess an electronic spin in the stable ground state! There may be one in a metastable state, which, however, seems to be extremely short-lived. So, it is unclear to me why the authors talk about long-lived spins while studying a system that does not seem to give access to them. While the material may have some nuclear spins, or spins of other paramagnetic impurities, it is totally unclear (at least to me) whether or how they could be accessed using the studied system.
- 2) The authors mention three central requirements for a scalable spin-photon interface: Long spin coherence, efficient spin-photon coupling, and telecom operation. The first are clearly out of reach for the system under study, and the telecom operation is only in the O-band. Other emitters, however, emit directly in the C-band, where loss in fibers is much lower (e.g. Dibos et al PRL 2018, Ourari et al. ArXiv 2022, Gritsch et al. ArXiv 2023), or can use efficient conversion (e.g. Dréau et al. Phys.Rev.App. 2018). So it is unclear to me why the studied system would be advantageous after all. Also, the claim that current platforms "fail to meet these requirements" is wrong, see e.g. Chen et al Science 2020, Ourari et al 2022 and Gritsch et al. 2023. In addition, there is another central requirement for quantum networking (that these systems seem to fulfill): Coherent photon emission. This topic is not touched upon in the manuscript, in spite of recent work (cited in the manuscript) that found that coherent emission will require enormous Purcell enhancement (Komza et al. Arxiv 2022), which, according to the manuscript under consideration, is extremely difficult to achieve because of the strong non-radiative decay.
- 3) The manuscript falls short of the state-of-the-art in many critical aspects. Other silicon cavities with coupled emitters routinely achieve Q-factors that are two orders of magnitude larger (Ourari et al, Gritsch et al), leading to 100-fold larger Purcell enhancement and up to 1000-fold lifetime reduction (Ourari et al). They also possess a spin, and emit in the telecom C-band, and

demonstrated coherent photon emission.

4) Eq (1) and (2) introduce the key figures of merit for quantum emitters in resonators, the outcoupling efficiency and the Purcell factor. However, none of these is actually determined in the manuscript, and it is difficult to find even in the supplement.

5) The paper claims to realize an efficient spin-photon interface. However, the count rates achieved e.g. in Komza et al. without a cavity, or in Redjem et al 2020, are much higher than in the studied devices. This is not mentioned in the main text, but only in the supplement on page 9, where ~ 1.5 kcts/s are reported, whereas the other experiments achieved ~ 10 kcts/s (which I only found when reading those references). In the main text, this is really hidden by reporting the counts in arbitrary units, and also in the comparison table at the end, the efficiency or count rate is not included. This is really strange; the authors mention several times that efficiency is a key figure of merit, and then this number cannot be extracted from the paper, even after trying for quite some time

6) The authors claim that "These challenges have so far resulted in weak and smallscale spin-photon coupling for current leading artificial atom platforms.". This is not true, consider e.g. Dibos et al, Ourari et al, Gritsch et al, Sipahigil et al Science 2016, Ulanowski et al Sci Adv 2022, and many more! All of these enable close-to-deterministic spin-photon coupling!

7) The authors mention that their emitter features a spin-triplet metastable state. However, the cited paper is theoretical, and I could not find any experimental evidence for such state. So, it should be made clear that this is expected, but may not be found after all. Also, judging from the discussion in the supplement, where different "types" of G-centers are compared, it seems to be very unclear what the nature of the emitter is, and if the observed emitter is identical to the one proposed in [18] or not (as ab-initio calculations never have the required accuracy to accurately predict transition frequencies...). In addition, in the supplement, the authors even present data that contradicts the metastable singlet hypothesis ("We attribute this difference to the fact that different emitters have different electron trapping rates and other mesoscopic properties that dictate the bunching." – all of this would not hold for a metastable state of the emitter)

8) The authors show data from two specific devices. However, it is not mentioned how many devices they had to measure before finding two that gave the desired results. However, this information would be critical to judge the scaling potential of the presented approach. In addition, the authors present only very few measurements. What is the reason for this? Is it that the devices degrade over time? If so, this should be mentioned. Otherwise, the reasons for the limited data set should be made clear.

9) The lack of data is most pronounced in Fig. 3. In a) only very few data points are measured to determine the dipolar dependence, where two tuning mechanisms were performed on two different devices. Why is it not measured on the same device, and why are there only two data points? To get a clear evidence for radiative enhancement, it would be required to find the functional form of the cavity in the data. With the two data points of the manuscript, also other mechanisms than cavity enhancement cannot be reliably discarded. As an example, gas condensation can change the surface properties and thus the carrier lifetime, and temperature tuning can change nonradiative decay mechanisms. Similarly, the optical mode and thus the emitter-to-detector coupling could change with T or with condensed gas. These effects would all show a different functional form than the cavity enhancement, so a thorough measurement could easily exclude them.

10) Many details of the experimental setup are missing or not clearly described, such that it is often not possible to extract what is actually measured, and how. This hinders reproducing the measurements. As an example, it is not mentioned if the HBT measurements use resonant or off-resonant excitation, and if the polarization plot in 3a uses linear or circular polarization in the excitation, or neither of them. Also, which measurements use which filtering setup is not mentioned explicitly. In addition, it is unclear if the data in 3f could be caused by different laser excitation conditions.

11) On several occasions, the data seems very preliminary, and not taken with the required care. As an example, it is mentioned in the supplement that the measurements were acquired "under likely different laser conditions, originating from an error related to our laser control electronics board." One would expect that paper data are collected after fixing such errors! Similarly, the data in the main figure are shifted by a random amount, as stated in the supplement: "which resulted in a slight wavelength offset likely due to calibration errors. This offset was taken into account when plotting the data, in order to enable a fair wavelength comparison [...]" One would expect that paper data is taken after a correct calibration of the instruments! Also, what does "a fair

comparison" mean here? In agreement with the expectation? How can the authors exclude that the system just degrades/changes over time, which causes deviations from the expectation? Maybe the calibration was correct, but the emitters shifted? This would also question the claimed efficiency change is due to the cavity tuning, rather than to a randomly fluctuating emitter.

12) In one of the cavities, the authors observe additional Fabry Perot resonances. In the other system, albeit designed identically, this is not observed. What is the reason for this? How can it be excluded that the emitter decays to both sides of the cavity, but the emission is channeled to the detector or away from it by these additional resonances of the setup? It seems that in Fig. 3e, the maximum and minimum emission coincides with maxima and minima of the FP mode. This may be a coincidence, but it is definitely an effect that will obscure the true enhancement of the emission via the resonator. Therefore, the claimed enhancement of the emission is very doubtful to me.

13) On page 2, the authors claim that their $g(2)$ value is "nearly an order of magnitude lower than the rest of the literature". This is heavily overclaiming and misleading. It ignores a huge amount of the literature, actually all emitters except the G-center, and also Hollenbach et al that is only a factor of 2 off with the same emitter.

14) The authors observe bunching with a 10 ns timescale, which they attribute to a metastable triplet state. This claim is highly doubtful. If it was such triplet state, why would it have a different amplitude in the two studied devices? Why would it have a different timescale, comparing to other works of the literature? How can the authors exclude spectral diffusion, another very common source of bunching in such measurements?

15) It seems that the cavities and emitters cannot be tuned reliably, but it is not mentioned what is causing this, and if this can be overcome. This may hinder reproducing the experimental results even when the same device is used! So, the authors should really explain this in detail.

16) From the Q/V ratio, formula (2), one would expect a Purcell factor of several hundred. Assuming that the dipole moment of the optical transition is reduced ~ 6 fold by the Debye Waller factor, and ~ 5 fold by the extracted quantum efficiency, one would still expect at least a tenfold lifetime reduction. This is in striking contradiction with the constant radiative lifetime! This deviation between theory and experiment is not mentioned in the text, not even in the supplement. To me, it again makes the claim that the observed enhancement of the fluorescence is due to the cavity enhancement very doubtful.

17) The authors claim that "The quantum efficiency (QE) of any silicon color center is one of the central unanswered questions in the field." I think that this is not the case – there are many works that investigate this, both for Erbium emitters and other color centers in silicon. However, the values obtained in the literature are often not compatible, which sheds some doubts in the used methods in this and other papers. So the authors should clarify why they think that their method is better than previous measurements, in particular in view of the mentioned weaknesses of their analysis.

18) In the discussion, the authors claim that "We show, for the first time, strong enhancement of the quantum emission of individual artificial atoms coupled to silicon nanocavities". This is again heavily overclaiming! Dibos et al and Ourari et al have measured almost thousandfold lifetime reductions! Redjem and Gritsch have also clearly demonstrated this. The authors mention these works after the manuscript as being published "during the preparation of this manuscript" – however, they were published on the ArXiv several months before the submission, so there would have been ample of time for an adequate discussion, and a comparison of the different works. In particular since the manuscript under consideration does not compete favorably, as it does not show a lifetime reduction that is the smoking gun of Purcell enhancement.

19) The authors claim in the outlook that "Our demonstration lays the groundwork for efficient spin-photon interfaces" – this is very bold, given that no efficiency is reported (and in the supplement one finds it is probably very lousy), and in addition the studied emitter does not even have a spin! In particular in view of other works that have clearly observed single spins in silicon with high efficiency (Higginbottom 2022, Gritsch 2023), it is completely unclear to me how the authors could arrive at such claims.

20) In the final part of the outlook the authors mention a collection of potential applications of their system. However, I cannot make sense of any of the mentioned topics, how the current work would relate to them! Ref 29 and 30 deal with photonic circuits that are programmable by temperature or electro-optic effects – I cannot see how the current work would add to this! Also, the mentioned photonic quantum information processing devices are clearly out of reach if the emitter is not coherent, which, following Komza et al, is very likely. Ref 34 is not even about deterministic entanglement generation, but about photonic cluster states. Ref 33 is about self-

learning algorithms – how would a G-center come in there? So, to summarize, the whole last paragraph does not seem to make any sense, and rather leaves the impression of a random collection of buzzwords.

21) The authors only mention in the methods that the fabrication of G-centers seems to unavoidably reduce the Q-factor of their nanophotonic devices. This leads to an inherently low efficiency. Why is this not mentioned and discussed in the main text, in particular since the whole paper is about the claim of enhancing efficiencies?

22) The authors claim that they achieve record-low $g(2)$, but they only do this on a single devices! Already the second one they study performs much worse. This should also be mentioned in the main text, instead of cherry-picking the best and then claiming any records.

23) In the supplement, they extract the lifetime from the $g(2)$ measurement. It is only 3.16 ns, so far from being consistent with the pulsed measurements (6 ns in 3f, 4.8ns in S3). This contradiction has to be resolved, in particular since the lifetime is the central element that goes into the QE analysis!

In addition to these main criticisms, I also have several smaller remarks that the authors should consider before resubmitting:

1) Fig 1 suggests that the authors have many identical devices on the same chip, but the actual measurements are performed only on two devices, so Fig. 1 is misleading.

2) I suggest to plot the spectrum in Fig. 1 on a log scale, such that the phonon sideband is visible. Otherwise, the reader gets the wrong impression that the Debye-Waller factor is close to unity

3) When the authors mention the first observation of spins in silicon, they should also include Chen et al. 2020 and Gritsch et al. 2023 that have measured single spins coupled to nanophotonic cavities.

4) The upper bound to the QE would urgently need an analysis of the statistical and systematic errors.

5) What is the yield of the G-center production mentioned in the methods?

6) Ref [6] gives a reference to the Purcell effect that seems of. The paper they want to cite was published in 1946 rather than 1995. Instead, I would suggest to pick a newer reference to a review paper that covers the Purcell effect, but also its realization with nanophotonic structures, e.g. Lodahl et al Rev.Mod.Phys. 2015

7) The authors mention in the supplement that the sample is mounted to a nanopositioning stage. It is a common issue to ensure thermalization in such setting in vacuum cryostats. So, the authors should mention how this is achieved, and where the temperature is measured in their experiments. If it is only the stage, the thermal tuning may have a completely wrong temperature range.

Reviewer #3 (Remarks to the Author):

The manuscript by Saggio et al. reports on the fabrication, characterization, and modeling of G-centers in silicon. Defect centers in silicon are attracting increasing attention as they combine the benefits of silicon photonics and silicon nanotechnology with localized quantum emitters, and the present manuscript reports on valuable additions to this nascent field. The main result is the realization of cavity-enhanced emission including the demonstration of a rather pure single-photon emission. The manuscript is generally clearly written and the figures are mostly readable and clear, but I would need clarification of a number of points before I can assess whether or not it is suitable for publication in Nature Communications. These points are – in order of importance:

1. The authors write under Notes that they have become aware of two recent papers on similar defects in cavities. This is not a proper way of discussing highly relevant and related works (one of which has since appeared as a journal publication: <https://www.nature.com/articles/s41467-023-38559-6>). Rather, the authors should discuss these works in the introduction and explain exactly at which points the present work advances the state of the art relative to these works. If the advances relative to previous works are not significant and clear, Nature Communications is hardly the right outlet.

2. In Eq. (1) as well as in the model for the quantum efficiency in the Supplementary Information, the authors seem to define the Purcell-enhanced rate as $(1+F_P)*\gamma_R$, where F_P is the Purcell factor and γ_R is the radiative rate of the transition. This definition is at least sloppy and perhaps also wrong: The Purcell-enhanced rate is $F_P*\gamma_R$, where γ_R is the radiative rate of an emitter in an infinitely extended medium with the same refractive index as the host material (in this case silicon). I do not understand the factor of one in this expression here and elsewhere in the manuscript. I would ask the authors to explain this model and also to be more rigorous regarding the difference between the radiative rate in the experiment and that in an infinitely extended medium.

3. For the cavity optimization, the authors refer to their previous work (Ref. 17) but the present manuscript needs to include more information on this as the present discussion is limited to Fig. S1. In particular, the authors should explain what is meant by the "0th Brillouin zone."

4. It can be tricky to make proper fits to decay curves, but only a single decay curve is shown in the Supplementary Information, and this is insufficient. The main text should include such raw data including the fit, both for the case of on- and off-resonance emission.

5. On a related note: Figure 3f is hard to read because the horizontal ticks are in deltas while the axis label advertises this as being in nanometers.

6. Figure 2 shows a number of figures including data on color scales – but without showing the color scale. This becomes particularly troublesome for Fig. 2d, which includes two unidentified color scales on top of each other. Surely, this can be plotted in a better way while showing all color scales.

7. The authors should include information about how many samples were characterized, if any data sets were discarded, etc. The Supplementary Information discusses these important points in too vague terms.

8. The reported quantum efficiencies in a number of related works are stated in the final table, but it appears not to be representative. In particular, Ref. 28 states that "The lifetime reduction and Purcell acceleration observed in our work for a single center indicates a close to unity quantum efficiency." While it is clear that this quantitative statement is hard to represent in a table, I am not convinced that leaving out this number when listing Ref. 28 is a fair representation of Ref. 28. If there are issues with the model used in Ref. 28 or if there is some other reasoning behind ignoring this statement, it should be explained in section 8 of the Supplementary Information.

Response to the Referee reports for manuscript NCOMMS-23-20767-T.

We thank the Reviewers for their helpful and constructive feedback. Below we give detailed point-by-point responses to all comments and questions.

.....

In response to Reviewer 1:

Reviewer 1: In the manuscript titled "Cavity-enhanced single artificial atoms in silicon," by V. Saggio et al., the authors present their research on cavity-enhanced single artificial atoms in silicon, which holds great potential for applications in quantum networks, scalable quantum computing, and sensing. The authors address the challenge of weak emission rates of these particular artificial atoms by coupling them to optical cavities and demonstrate the optimization of photonic crystal cavities for controllable cavity-coupling of single G-centers operating in the telecommunications O-band.

Silicon as a host material for single artificial atoms operating in the telecommunication bands has a unique potential to combine long spin coherence times with telecommunication wavelength photons, leveraging at the same time the success of silicon microelectronics and photonics. This makes the silicon platform of particular interest to the quantum photonics community.

The weak light-matter interaction leads to a low single-photon emission rate by quantum emitters in solids, which is typically below 1 GHz. Coupling of single-photon emitters to optical resonators allows to substantially enhance the light-matter interaction and speed up the emission rate. In this work, the authors address this challenge by coupling G-centers in silicon to 2D photonic crystal cavities. Notably, the authors apply the design approach to achieve high-quality cavities that offer both high Purcell enhancement, Q/V (quality factor per mode volume), and high coupling efficiency η . This not only enhances the single-photon generation but also ensures high collection efficiency, which both constitute the key performance metric of the single-photon source.

The paper provides experimental evidence for the successful coupling of single artificial atoms to the optimized cavities. The measurements confirm the presence of single G-centers in the photonic crystal cavities and demonstrate an enhancement of the emitter's single-photon emission.

Therefore, I judge this manuscript to be of interest to the Nature Communications readership and quantum information research community. However, before I can recommend publication, I have several questions (listed below) that must be addressed.

Our reply: We thank the Reviewer for the overall positive assessment. Below we address their concerns in a detailed point-by-point response.

.....

Reviewer 1: 1. How would the authors explain no change in the excited state lifetime? Clear lifetime shortening was reported in Redjem, W. et al. All-silicon quantum light source by embedding an atomic emissive center in a nanophotonic cavity. arXiv preprint arXiv:2301.06654 (2023)

Our reply: The observed no change in lifetime can be explained with the presence of non-radiative decay channels. If their effects are predominant on radiative decay mechanisms, the effectiveness of the expected Purcell enhancement would be reduced. In our experiment, we attribute this mainly to the presence of a metastable state, i.e. level 3 shown in Fig. 1c. When the excited state (level 2) is populated, the emitter can either decay to the ground state (level 1) through radiative processes, that is emitting photons, or to level 3 through non-radiative processes. If the latter effects are large enough, the effective Purcell enhancement would be shadowed by the non-radiative decay rate, and no significant change in the excited state lifetime would be observed. Our observation matches what recently reported in [Lefaucher, B. et al. *Appl. Phys. Lett.* **122**, 061109 (2023)] for ensembles of G-centers in silicon.

Moreover, the Reviewer is correct that clear lifetime shortening was recently reported in [Redjem, W. et al. *arXiv:2301.06654* (2023)]. We attribute this disparity to the possibility that a different carbon-related defect is being considered in their work. As we discuss in the SI, Sec. 8, their observed or estimated values of ZPL wavelength, inhomogeneous distribution, quantum efficiency and lifetime do not match the rest of the literature for G-centers. Recently, the possibility of two different G-center-like structures was brought forward by [Baron, Y. et al. *Appl. Phys. Lett.* **121**, 084003 (2022)]. We believe that our work corroborates this hypothesis, although the mechanisms leading to the formation of different defect configurations is still an open question and requires further investigation.

We are thankful to the Reviewer for raising this point, which showed us that further clarification is needed in our manuscript. We have now added the following to the main text Discussion:

*“In our experiments however, no significant change in the excited state lifetime is observed. These results align with what recently reported for ensembles of G-centers [Lefaucher, B. et al. *Appl. Phys. Lett.* **122**, 061109 (2023)]. In general, potential angular and spatial mismatch between the emitter’s dipole moment and the cavity mode electric field would contribute to reduce the Purcell effect. However, we attribute the observance of constant lifetime mainly to the presence of strong non-radiative decay processes, which compete with radiative decay and thus effectively reduce the Purcell enhancement. Such non-radiative processes may be attributed to the presence of a metastable state in the G-center configuration, where the atom can decay to without emitting photons. In principle, achieving higher quality factors (see Methods, Sec. B), smaller mode volumes, and better cavity-emitter spatial alignment may induce enhancement that could outweigh non-radiative decay effects, thus resulting in lifetime reduction. This may also lead to a system with higher coherent photon emission, highly desirable for quantum information processing [Komza, L. et al. *arXiv:2211.09305* (2022)]. Lifetime shortening down to ~ 7 ns — comparable to our measured values even in the absence of lifetime reduction — was recently observed in Ref. [Redjem, W. et al. *Nat. Commun.* **14**, 3321 (2023)] for supposedly the same system. However, a hypothesis was recently raised regarding the possibility of two different physical systems being reported as G-centers [Baron, Y. et al.*

Reviewer 1: 2. Along those same lines, how do the saturation measurements of the emitter compare when it is in resonance vs when it is detuned from the cavity? In Figure 3f, the authors show an increase in counts on-resonance vs off-resonance, but these numbers are in arbitrary units, and much lower than the saturation measurement shown in Figure 3b. I would suggest that saturation be compared at different cavity detuning since the coupled emitter should have a high saturated intensity than when it is uncoupled.

Our reply: We thank the Reviewer for raising this point, which shows us that we need to further clarify Fig. 3. The saturation measurements shown in Fig. 3b are taken when the emitter is closely coupled to the cavity. Although we do not have saturation measurements at different detunings, we clarify that the difference in counts compared to Fig. 3f is due to the use of a different spectrometer grating density. A density of 300 gr/mm was used in Fig. 3b, while the gas-tuned data in Fig. 3f was acquired with a density of 900 gr/mm. The varying grating efficiencies and spectrometer pixel projections result in an intensity difference between the two cases. However, this is just an offset and does not affect the value of the saturation power. We now clarify this in the main text by stating the following:

“The gas tuning data in Fig. 3f were acquired with a different spectrometer grating density compared to the data reported in Fig. 3b, hence showing lower counts compared to the saturation measurements (see SI, Sec. 4).”

Reviewer 1: 3. What limits the quality factor of the fabricated photonic crystal cavities? The authors predict Q-factors of up to 5.0×10^5 theoretically but only measure $Q \sim 5000$.

Our reply: We attribute this disparity to the high carbon doping density that is needed to produce cavity-coupled G-centers. We discuss this in more detail in Sec. B of the Methods, where we also suggest that reducing the density or applying localized doping could improve the cavities’ quality factors as follows:

*“As already mentioned in the main text, we observe quality factors much lower than the $Q \sim \mathcal{O}(10^6)$ result expected from both simulation (Fig. S3) and previous statistical studies on thousands of photonic crystals designed for ~ 1550 nm operation under the same optimization method [Panuski, C.L. et al. *Nat. Photon.* **16**, 834-842 (2022)]. We attribute this disparity to the high carbon doping density used to produce cavity-coupled G-centers with sufficient probability. Reducing the doping density or applying localized doping [Hollenbach, M. et al. *Opt. Express* **28**, 26111–26121 (2020)] could play a role in recovering performance closer to intrinsic silicon. Applying large-scale characterization techniques [Sutula, M. et al. *Nature Materials* 1-7 (2023)] to locate ideal emitters and*

fabricate cavities around these positions could enhance the yield of coupled emitters in the case of reduced doping density.”

.....

Reviewer 1: 4. How is the emitter dipole oriented relative to the cavity?

Our reply: In our experiment, we only observe emission from G-centers that spectrally overlap with the cavity profile to some degree. In other words, we are filtering out the emission that is not coupled to the cavity, and therefore observe only the portion of emission that overlaps with the cavity mode. The orientation of the emitter dipole has been measured and can be found in previous works such as [Lefaucher, B. et al. *Appl. Phys. Lett.* **122**, 061109 (2023), Redjem, W. et al. *Nat. Electron.* **3**, 738–743 (2020)].

.....

Reviewer 1: 5. What is the fabrication yield of the devices? Based on the presented example as the best result, what potential approaches do the authors suggest for improving performance and achieving greater scalability that could be explored immediately?

Our reply: We thank the Reviewer for raising this point, which we are happy to address. In order to make statistically sound conclusions about distributions and yields, large datasets would be necessary. We are working to extend our automated spectroscopy techniques [Sutula, M. et al. *Nature Materials* 1-7 (2023)] to the IR regime, but face challenges from IR detection and lower signals as explained below. While preliminary, these studies provide important insights despite limitations on statistical claims. The challenges for automated big-data spectroscopy for us presently are as follows, which makes estimating the yield challenging. First of all, our chip hosts thousands of cavities which differ in size and quality factor, and feature resonance wavelengths spanning a relatively wide range (from ~ 1270 to ~ 1320 nm). Our fabrication approach involves individually modeling each of our cavities through an inverse design process. This means that we do not replicate identical cavity designs; rather, each cavity will differ from all others making it challenging to extract the yield. The difference in cavity parameters results in most emitters being decoupled, with some cavities featuring potentially too low Q to show enhancement or far off resonance wavelength, or with emitters not being spatially aligned to cavities. Moreover, with full decoupling, our setup cannot detect ZPL emission, obscuring yield estimates. A first approach to improve scalability could be targeting the fabrication of cavities with a resonance wavelength closer to the ZPL of our emitters, thus rendering the tuning process easier to achieve. Other tuning mechanisms such as tuning via electric fields [Anderson, C.P. et al. *Science* **366**, 1225 (2019)] or mechanical strain [Wan, N.H. et al. *Nature* **583**, 226 (2020)] may be applied as well to ensure spectral alignment of emitter and cavity. On the other hand, due to the intrinsically random carbon implantation process, we deal with probabilistic creation of G-centers in our cavities. A viable way to improve this may be the deterministic creation of single G-centers in cavities using techniques such as femtosecond laser annealing [Quard, H. et

al. *arXiv:2304.03551* (2023)] or focused ion beam [Hollenbach, M. et al. *Nat. Commun.* **13**, 7683 (2022)], which would lead to the creation of G-centers in predefined locations, enabling scalability. We discuss all of this in the main text Discussion by stating the following:

*“A central requirement for the scalability of our system is localized spatial and spectral alignment of both many cavities and many atoms to a common global frequency. The spatial alignment of the cavity and atom can be achieved by making use of the recently reported localized implantation of single G- and W-centers [Hollenbach, M. et al. *Nat. Commun.* **13**, 7683 (2022)]. Silicon artificial atoms can be spectrally aligned using the recently reported non-volatile optical tuning for G-centers [Prabhu, M. et al. *Nat. Commun.* **14**, 2380 (2023)] or methods used in other artificial atom systems such as tuning via electric fields [Anderson, C.P. et al. *Science* **366**, 1225 (2019)], or mechanical strain [Wan, N.H. et al. *Nature* **583**, 226 (2020)]. Cavity tuning via local thermal oxidation of silicon has been achieved on a large scale [Panuski, C.L. et al. *Nat. Photon.* **16**, 834-842 (2022)], and a similar method could be used to align large arrays of cavity-atom systems at room or cryogenic temperatures.”*

.....

Reviewer 1: 6. The argument the authors make that this work is conclusive evidence of two closely related but different types of G-centers is somewhat lacking in evidence. For example, the authors claim that Redjem et al show lifetime shortening of a different type of defect center since they report an uncoupled excited state lifetime of 30 ns. However, Redjem et al report a ZPL wavelength of 1275 nm, which is close to the 1279 nm ZPL wavelength reported here; the discrepancy between the two could be explained by variations in strain originating from damage induced by implantation. The possibility of different color centers in Si due to carbon implantation should be investigated more systematically before such a claim can be made.

Our reply: We thank the Reviewer for raising this point. While we do observe two clearly distinct clusters (differing not only in ZPL wavelength but also in homogeneous distribution, lifetime and quantum efficiency), and thus hypothesize that two different types of G-centers are likely being studied, we agree with the Reviewer that a more thorough investigation — beyond the scope of this work — would be needed. We already state that these reported differences might be due to e.g. a different host material, fabrication protocols etc in the SI, Sec. 8. However, we now make this also clear in the main text Discussion as follows:

*“However, a hypothesis was recently raised regarding the possibility of two different physical systems being reported as G-centers [Baron et al. *Appl. Phys. Lett.* **121**, 084003 (2022)]. Table I in SI, Sec. 8 compares the reported experimental results for single G-center labeled artificial atoms in silicon, ...*

...

More information on this comparison can be found in SI, Sec. 8. While our findings hint at the potential existence of two distinct artificial atom systems, the observed variations could still stem from differences in measurement configurations, strain, or fabrica-

tion methods. We thus conclude that our work highlights the need for further theoretical and experimental investigation regarding the creation process and the photophysics of G-center-like artificial atoms in silicon platforms.”

.....

Reviewer 1: Nevertheless, this work is an important step in silicon-based quantum photonics using artificial atoms, and I judge it likely to be of broad interest. I recommend major revision as the best option.

Our reply: We thank the Reviewer for the constructive feedback, which have definitely helped us improve our work. In light of the changes we have made to address all the Reviewer’s concerns, we now hope that our work will be found suitable for publication.

In response to Reviewer 2:

Reviewer 2: Saggio et al. report an experimental study on single-photon emitters that are embedded into a nanophotonic resonator made from silicon. The study is put into the context of distributed quantum information processing, or quantum networking, which is a topic that is of high relevance to a broad audience. Compared to other materials under study, such as quantum dots in GaAs or color centers in diamond or silicon carbide, the presented choice of emitter and host has several advantages: First, nanofabrication in silicon is a well-advanced technology, which may aid the up-scaling of corresponding devices. Second, emission into the telecom wavelength band allows for transmission over optical fibers with reduced loss. Albeit these improvements seem technical at first sight, they pose critical steps towards up-scaling of quantum technologies. Thus, the subject matter is in principle suited for publication in Nature Communications. However, the manuscript in the current form is not. The reason is that the achievements of the authors are heavily oversold, and the central claims of the paper are not backed by the measurements. Even worse, much of the information that is required to judge the achievements (or the shortcomings) of the manuscript is only accessible after studying the SI in detail, while it does not show up in the main text.

Our reply: We thank the Reviewer for the detailed feedback. We are happy to address the raised concerns in what follows.

.....

Reviewer 2: In detail, the main criticisms are: 1) The paper is put into the context of quantum networking. The authors mention several times that this requires long-lived spins. Thus, one would expect that the experimental data includes some evidence for such long-lived spins. However, there is no corresponding measurement. Even after studying the supplement, I could not tell whether such spin is present or even expected in the G-center. Only after reading some of the references, I understood that the studied emitter does not possess an electronic spin in the stable ground state! There may be one in a metastable state, which, however, seems to be extremely short-lived. So, it is unclear to me why the authors talk about long-lived spins while studying a system that does not seem to give access to them. While the material may have some nuclear spins, or spins of other paramagnetic impurities, it is totally unclear (at least to me) whether or how they could be accessed using the studied system.

Our reply: We thank the Reviewer for their feedback, and we are happy to expand on this point. In light of this and all other raised comments, we have substantially modified our Introduction, which now focuses on the advantages of color centers in silicon and provides a clear comparison between our work and the rest of the literature. We now make it more clear that our studied defect might not possess an addressable spin, or that at least this has not been investigated for single G-centers yet. We refer the Reviewer to point 7 of this response, which further elaborates on this matter in great detail.

.....

Reviewer 2: 2) The authors mention three central requirements for a scalable spin-photon interface: Long spin coherence, efficient spin-photon coupling, and telecom operation. The first are clearly out of reach for the system under study, and the telecom operation is only in the O-band. Other emitters, however, emit directly in the C-band, where loss in fibers is much lower (e.g. Dibos et al PRL 2018, Ourari et al. ArXiv 2022, Gritsch et al. ArXiv 2023), or can use efficient conversion (e.g. Dréau et al. Phys.Rev.App. 2018). So it is unclear to me why the studied system would be advantageous after all.

Our reply: We thank the Reviewer for bringing this to our attention. This shows us that further clarification about the current state-of-the-art of emitters in silicon is needed. In our manuscript, we are not claiming the implementation of a spin-photon interface. Instead, we introduce artificial atoms in silicon as promising candidates for quantum information processing, and subsequently focus our discussion on G-centers.

These color centers emit in the O-band, which exhibits losses of up to 0.4 dB/km in silica optics. We agree with the Reviewer that the C-band provides even lower losses (~ 0.25 dB/km), however this difference is not large enough to exclude the O-band from being well-suited for long-distance information transfer. Moreover, while it is true that other emitters [Dibos, A.M. et al. *Phys. Rev. Lett.* **120**, 243601 (2018), Ourari, S. et al. *arXiv:2301.03564* (2023), Gritsch, A. et al. *Optica* **10**, 783-789 (2023), Berkman, I.R. et al. *Phys. Rev. Appl.* **19**, 014037 (2023)] emit directly into the C-band, all the cited papers feature the investigation of erbium dopants. The lifetime of such systems is very long (in the order of \sim ms in the bulk) and Purcell enhancement has been demonstrated that reduces it down to \sim μ s. This is still three orders of magnitude higher than what we observe for G-centers, despite the fabrication of cavities with high quality factors in the aforementioned works. The intrinsically long lifetime unavoidably leads to low photon emission rates. As discussed in [Dibos, A.M. et al. *Phys. Rev. Lett.* **120**, 243601 (2018)], enormous Purcell factors ($> 10^5$) would be required to enhance the photon emission to 10 MHz, since the initial rate is only in the order of tens of Hz. Similarly, in [Berkman, I.R. et al. *Phys. Rev. Appl.* **19**, 014037 (2023)] they estimate that a Purcell factor of approximately 2×10^6 would be needed to achieve emission rates of 400 MHz. However, such large Purcell factors have not been demonstrated yet, despite the use of cavities with very high quality factors.

We would also like to briefly comment on the suggested efficient frequency conversion solutions. Despite the potentially high conversion efficiency (which is still however limited by several factors), realizing such a conversion system requires a notable overhead in resources, e.g. additional pumps and crystals, and has to deal with added noise and additional coupling losses. This overhead would be totally avoided in the case of color centers emitting directly into the telecom band, which constitutes a clear advantage.

We are thankful to the Reviewer for this comment, which has shown us that we need to make a clearer distinction between emitters in silicon and discuss their trade-offs. We have extensively revised the Introduction to clarify the context, emphasize the advantages of G-centers in silicon, and compare to other emitters. As the changes are quite substantial, we refer the Reviewer directly to the first half of the Introduction in the revised manuscript.

.....

Reviewer 2: Also, the claim that current platforms “fail to meet these requirements” is wrong, see e.g. Chen et al Science 2020, Ourari et al 2022 and Gritsch et al. 2023. In addition, there is another central requirement for quantum networking (that these systems seem to fulfill): Coherent photon emission. This topic is not touched upon in the manuscript, in spite of recent work (cited in the manuscript) that found that coherent emission will require enormous Purcell enhancement (Komza et al. Arxiv 2022), which, according to the manuscript under consideration, is extremely difficult to achieve because of the strong non-radiative decay.

Our reply: The works cited by the Reviewer refer again to the use of single erbium dopants. In this sense, despite the successful demonstration of spin-photon coupling, such systems face their own challenges due to the very low count rates even under Purcell enhancement.

Moreover, we thank the Reviewer for bringing the concern about coherent photon emission to our attention, and we acknowledge that this aspect should be properly discussed in our manuscript. While it is true that [Komza, L. et al. *arXiv:2211.09305* (2022)] discuss ways to improve the coherence, they state that mode volumes $< 0.1\lambda^3$ and quality factors of $\sim 10^6$ could Purcell enhance the coherence atom-photon interaction rate by $\sim 10^6$. These numbers have been already demonstrated in our previous work [Panuski, C.L. et al. *Nat. Photon.* **16**, 834-842 (2022)]. As we point out in our manuscript, the carbon implantation process reduces the quality factors (from the expected 10^6 , achieved with the same method described in [Panuski, C.L. et al. *Nat. Photon.* **16**, 834-842 (2022)] to 10^3). However, we already discuss approaches to improve or avoid this problem in the Methods, Sec. B. While we assume that strong non-radiative decay is the main factor shadowing our Purcell enhancement, it cannot be excluded that higher quality factors and smaller mode volumes could still reveal enhancement that could potentially outweigh non-radiative decay effects. In light of all of this, the prospects for achieving high coherence from G-centers remain an open possibility. We now make these points clear in the Introduction as well as in the main text Discussion by stating the following:

*“In principle, achieving higher quality factors (see Methods, Sec. B), smaller mode volumes, and better cavity-emitter spatial alignment may induce enhancement that could outweigh non-radiative decay effects, thus resulting in lifetime reduction. This may also lead to a system with higher coherent photon emission, highly desirable for quantum information processing [Komza, L. et al. *arXiv:2211.09305* (2022)].”*

.....

Reviewer 2: 3) The manuscript falls short of the state-of-the-art in many critical aspects. Other silicon cavities with coupled emitters routinely achieve Q-factors that are two orders of magnitude larger (Ourari et al, Gritsch et al), leading to 100-fold larger Purcell enhancement and up to 1000-fold lifetime reduction (Ourari et al). They also possess a spin, and emit in the telecom C-band, and demonstrated coherent photon emission.

Our reply: As already discussed above, the papers cited by the Reviewer refer to the use of erbium dopants in silicon. While it is true that high quality factors and consequently high Purcell enhancement have been demonstrated, the lifetime has been shortened down to $\sim \mu\text{s}$, which is still three orders of magnitude larger than what reported for G-centers, even in the absence of lifetime reduction. We have provided a detailed discussion about this matter in the preceding points and modified the Introduction accordingly.

.....

Reviewer 2: 4) Eq (1) and (2) introduce the key figures of merit for quantum emitters in resonators, the outcoupling efficiency and the Purcell factor. However, none of these is actually determined in the manuscript, and it is difficult to find even in the supplement.

Our reply: We agree with the Reviewer’s observation that additional clarification is required in this regard. The outcoupling efficiency (70%) is actually already reported both in the main text and SI. Additionally, we now refer to Eq. 2 and derive an estimate of the Purcell factor F_P (a few hundreds). We comment more on what would be expected versus what we observe in the main text Discussion as follows:

“From Eq. 2, assuming $Q \sim 3000$ and $V \sim (\lambda/n)^3$, we can estimate a Purcell factor F_P of a few hundreds, which would be expected to lead to a perceivable lifetime reduction. In our experiments however, no significant change in the excited state lifetime is observed.”

More details about this are provided in point 16 of this response.

.....

Reviewer 2: 5) The paper claims to realize an efficient spin-photon interface. However, the count rates achieved e.g. in Komza et al. without a cavity, or in Redjem et al 2020, are much higher than in the studied devices. This is not mentioned in the main text, but only in the supplement on page 9, where 1.5 kcts/s are reported, whereas the other experiments achieved 10 kcts/s (which I only found when reading those references). In the main text, this is really hidden by reporting the counts in arbitrary units, and also in the comparison table at the end, the efficiency or count rate is not included. This is really strange; the authors mention several times that efficiency is a key figure of merit, and then this number cannot be extracted from the paper, even after trying for quite some time

Our reply: We thank the Reviewer for bringing this crucial point to our attention, and we are happy to provide clarification on this matter. The count rates achieved in previous works are actually not higher than what we measure. This is because our saturation measurements were taken extracting counts at different optical powers from measured spectra, and not using our single-photon detectors. We acknowledge that this should have been made more clear. Measuring counts of 93 kcounts/min with our spectrometer corresponds to observing ~ 40 kcounts/s with our SNSPDs. Correcting for the relatively low efficiency of our detectors ($\sim 20\%$), we would therefore observe

numbers in the order of 200 kcounts/s. Although a direct comparison with other works is not straightforward due to the different experimental parameters (e.g. the N.A. of the objective or filtering apparatus, which drastically impact the collection efficiency), we can qualitatively estimate the rates taking into account the detector efficiencies. For example, in [Komza, L. et al. *arXiv:2211.09305* (2022)], they extract a rate of ~ 35 kcounts/s using detectors with efficiency of $\sim 60\%$. Correcting for this efficiency, they would measure ~ 60 kcounts/s. Similarly, [Hollenbach, M. et al. *Nat. Commun.* **13**, 7683 (2022)] observe ~ 13 kcounts/s, which is close to what they would measure in the case of perfect detection since they use detectors with $\sim 90\%$ efficiency. [Hollenbach, M. et al. *Opt. Express* **28**, 26111–26121 (2020)] observe ~ 14 and ~ 99 kcounts/s with the same detection efficiency. [Prabhu, M. et al. *Nat. Commun.* **14**, 2380 (2023)] measure instead ~ 5 kcounts/s, which would correspond to ~ 22 kcounts/s taking into account the $\sim 23\%$ detection efficiency. [Redjem, W. et al. *Nat. Electron.* **3**, 738–743 (2020)] observe ~ 8 kcounts/s at saturation, which would correspond to ~ 80 kcounts/s considering the $\sim 10\%$ detection efficiency. [Redjem, W. et al. *Nat. Commun.* **14**, 3321 (2023)] measure ~ 20 kcounts/s in the on-resonance case, however it is unclear what the detection efficiency is and how this aligns with their previous work. We highlight that the last two works feature a higher objective N.A. compared to our case. Moreover, it is possible that they both consider a different type of G-center, as discussed in the last section of the SI. Even though this analysis is purely qualitative, the efficiency reported in our work seems to be much higher than previous works on G-centers. However, since we acknowledge that more experimental details would be needed to compare all the collection efficiencies in a fair way, we decided not to include this comparison in the table. We nevertheless discuss this now both in the main text and SI as follows:

“We highlight here that the counts reported in the saturation curve are extracted from spectroscopy measurements. The corresponding intensity value measured with our SNSPDs at saturation is ~ 40 kcounts/s. Although a direct comparison with other works on G-centers is not straightforward due to different experimental parameters, we can conclude that our emitter features a notably high single-photon count rate (see SI, Sec. 8).”

“Even though a direct comparison of the efficiencies among these works is challenging due to different experimental parameters such as N.A. of the objective, filtering apparatus, and coupling and detection efficiency, we can qualitatively correct for the detection efficiency, when reported, and extract an indicative value of the count rates at saturation. Most of the cited works reporting saturation measurements on single G-centers would measure rates well below ~ 100 kcounts/s. In our case, we would measure ~ 200 kcounts/s instead. While this is just a qualitative estimate, we can conclude that our ZPL enhancement leads to notably bright single-photon emission.”

.....

Reviewer 2: 6) The authors claim that “These challenges have so far resulted in weak and smallscale spin-photon coupling for current leading artificial atom platforms.”. This is not true, consider e.g. Dibos et al, Ourari et al, Gritsch et al, Sipahigil et al Science 2016, Ulanowski et al Sci Adv 2022, and many more! All of these enable close-to-deterministic spin-photon coupling!

Our reply: In light of the previous points, a substantial part of the Introduction has been

rewritten to clarify the state-of-the-art of silicon quantum emitters, and does no longer contain this statement. We now start with discussing quantum emitters in silicon, and then focus on color centers, for which spin-photon coupling has not been demonstrated yet. We are thankful to the Reviewer for their comments, which have definitely helped us improve the overall clarity of our Introduction.

.....

Reviewer 2: 7) The authors mention that their emitter features a spin-triplet metastable state. However, the cited paper is theoretical, and I could not find any experimental evidence for such state. So, it should be made clear that this is expected, but may not be found after all. Also, judging from the discussion in the supplement, where different “types” of G-centers are compared, it seems to be very unclear what the nature of the emitter is, and if the observed emitter is identical to the one proposed in [18] or not (as ab-initio calculations never have the required accuracy to accurately predict transition frequencies. . .). In addition, in the supplement, the authors even present data that contradicts the metastable singlet hypothesis (“We attribute this difference to the fact that different emitters have different electron trapping rates and other mesoscopic properties that dictate the bunching.” – all of this would not hold for a metastable state of the emitter)

Our reply: The Reviewer is correct that the paper cited to claim the presence of a metastable state is theoretical. A stronger evidence for such a state may come from [Lee, K.M. et al. *Phys. Rev. Lett.* **48**, 37 (1982)], where optically detected magnetic resonance (ODMR) revealed the presence of a spin triplet state in ensembles of G-centers. However, ODMR has so far not been replicated for single G-centers. For this reason, we now cite this experimental paper and make it clear that a spin triplet metastable state is expected in G-centers but has yet to be confirmed:

*“The G-center is a quantum emitter formed by two substitutional carbon atoms and a silicon interstitial (Fig. 1b), and features a zero phonon line (ZPL) transition at 970 meV (1279 nm) in the telecommunications O-band along with an expected spin triplet metastable state [Udvarhelyi, P. et al. *Phys. Rev. Lett.* **127**, 196402 (2021), Lee, K. et al. *Phys. Rev. Lett.* **48**, 37-40 (1982)], which has so far been observed in ensembles only (Figs. 1c, d).”*

Moreover, the observed properties of the emitter are in line with other works focusing on addressing single G-centers (see SI, Sec. 8). Therefore, our conclusions are not only based on the cited theoretical paper [Udvarhelyi, P. et al. *Phys. Rev. Lett.* **127**, 196402 (2021)] but also on all other works reporting similar properties for G-centers. We further discuss the possible presence of a metastable state in the SI, Sec. 5 as well. There, we state that the bunching may be linked to the presence of such a state. The two different $g^{(2)}(t)$ amplitudes cannot be directly compared because of the different optical powers used, and are therefore not enough to rule out the possible presence of this third state. We now make this point more clear in our manuscript (see point 14 of this response, where we address this aspect in detail).

.....

Reviewer 2: 8) The authors show data from two specific devices. However, it is not mentioned how many devices they had to measure before finding two that gave the desired results. However, this information would be critical to judge the scaling potential of the presented approach.

Our reply: We agree with the Reviewer that this point needs further clarification. Our sample hosts thousands of cavities, fabricated in different sizes, spanning a relatively wide resonance wavelength range (from ~ 1270 to ~ 1320 nm), and optimized for different parameters such as the Q-factor and the far-field profile. Our fabrication approach involves individually modeling each of our cavities through an inverse design process. This means that we do not replicate identical cavity designs; rather, each cavity will differ from the others. In our study, only one sample was used and a few cavities (~ 10) were targeted based on the Q-factor, size and resonance wavelength. Each cavity was then probed to look for the presence of a single emitter. Due to the intrinsically random carbon implantation process, not every cavity will feature the presence of a single emitter (we already discuss methods to improve this in the main text Discussion.) In our work, only one sample was used and the discarded data sets were the ones that did not show the presence of an emitter, or that showed multiple ones. As the Reviewer suggests, we now discuss this point in the SI, Sec. 2:

“In our experiment, one sample hosting thousands of cavities — each nominally different from all others — was used. After targeting a few cavities (~ 10) based on Q-factor, size and resonance wavelength, each of them was probed to look for the presence of a single emitter. Two of them were then chosen and analyzed. We discarded data sets that did not show the presence of an emitter, or that revealed multiple ones.”

.....

Reviewer 2: In addition, the authors present only very few measurements. What is the reason for this? Is it that the devices degrade over time? If so, this should be mentioned. Otherwise, the reasons for the limited data set should be made clear. 9) The lack of data is most pronounced in Fig. 3. In a) only very few data points are measured to determine the dipolar dependence, where two tuning mechanisms were performed on two different devices. Why is it not measured on the same device, and why are there only two data points? To get a clear evidence for radiative enhancement, it would be required to find the functional form of the cavity in the data. With the two data points of the manuscript, also other mechanisms than cavity enhancement cannot be reliably discarded. As an example, gas condensation can change the surface properties and thus the carrier lifetime, and temperature tuning can change nonradiative decay mechanisms. Similarly, the optical mode and thus the emitter-to-detector coupling could change with T or with condensed gas. These effects would all show a different functional form than the cavity enhancement, so a thorough measurement could easily exclude them.

Our reply: We are thankful to the Reviewer for raising this point, which has made us realize that a more detailed analysis of the cavity-atom coupling mechanisms is necessary. We now present additional measurements, which show how this coupling is affected by

the tuning. While the device is very stable and does not degrade over time, the ZPL may shift when tuning the cavity resonance (as we already mention in the SI). This is a known effect already observed in [Prabhu, M. et al. *Nat. Commun.* **14**, 2380 (2023)]. We now discuss this in great detail in Sec. 6b of the SI, where we provide additional plots showing intermediate tuning steps. In short, while the cavity blueshifts as a consequence of gas sublimation, the ZPL spectrally shifts as well. As the maximum magnitude of this spectral shift has been measured to be around 0.3 nm [Prabhu, M. et al. *Nat. Commun.* **14**, 2380 (2023)], this effect results in the need for additional tuning steps to achieve significant decoupling. Potential mechanisms explaining this ZPL shift are to be found in [Prabhu, M. et al. *Nat. Commun.* **14**, 2380 (2023)]. Alternative ways to (de)couple cavities and emitters such as tuning via electric fields or mechanical strain are already mentioned in the Discussion. However, we now mention this in the SI as well. We add to Sec. 6 as follows:

*“Achieving the detuning δ'_g starting from δ_g is the result of several intermediate tuning steps, each corresponding to a different laser power used to tune the cavity. In more detail, we indicate with step 0 the initial situation where no tuning is performed, and with step 1 to step 5 the stages where powers of 250 μW , 320 μW , 400 μW , 450 μW and 550 μW , respectively, were used to burn the gas off the surface of the sample. Fig. S9b shows step 0 (corresponding to δ_g) and step 5 (corresponding to δ'_g) only. As mentioned above, this tuning mechanism blueshifts the cavity resonance, but may also spectrally shift the ZPL [Prabhu, M. et al. *Nat. Commun.* **14**, 2380 (2023)]. Examples of ZPL shifts are shown in Fig. S11a for the different tuning steps. From Lorentzian fits, we extract central wavelengths of (1279.277 \pm 0.001) nm, (1279.239 \pm 0.001) nm, (1279.167 \pm 0.001) nm, (1279.0988 \pm 0.0002) nm, (1279.126 \pm 0.002) nm, (1279.4587 \pm 0.0003) nm for steps 0 to 5, respectively. In practice, these shifts affect the coupling of the emitter to the cavity, whose resonance always blueshifts at each step. Fitting the cavity data with the same procedure described above, we extract central wavelengths of (1279.354 \pm 0.002) nm, (1279.316 \pm 0.001) nm, (1279.290 \pm 0.001) nm, (1279.211 \pm 0.002) nm, (1279.137 \pm 0.002) nm, (1278.976 \pm 0.001) nm for steps 0 to 5, respectively. Fig. S11b shows the counts in function of the relative shift between the cavity resonance and the ZPL wavelength. From step 0 to 4, this shift stays small, which results in the observation of relatively similar counts (with step 4 showing maximum coupling). One would expect a monotonic behaviour when increasing the relative spectral shift. However, small variations in the intensity may be attributed to the FP oscillations, which may play a role in enhancing or de-enhancing the emission. The FP cavity profiles and ZPL spectra for each tuning step are reported in Fig. S12. There, it is visible that the emission at e.g. step 0 corresponds to a dip in the cavity profile. This means that, in principle, one would expect higher ZPL intensity in the ideal case without FP effects. Similarly, the emission in step 2 coincides with a maximum FP oscillation, meaning that its intensity would be lower in the ideal case. Similar conclusions can be drawn about all other steps. This would explain the non-monotonic behaviour of the counts in Fig. S11b. Additionally, beam repositioning after each tuning step might introduce minor additional errors to the measurements. At step 5, the cavity and emitter shift significantly in opposite directions, thereby effectively reducing their coupling. This results in a significant decrease in the ZPL intensity, as visible in Fig. S11b. As discussed in the main text, alternative approaches to (de)couple the emitter from the cavity may be based on electric field tuning [Anderson, C. P. et al. *Science* **366**, 1225–1230 (2019)] or mechanical strain [Wan, N. H. et al. *Nature* **583**, 226–231 (2020)].”*

The newly provided data can be found in the manuscript. Additionally, we now highlight the ZPL shift in the main text as well by stating the following:

“... we observe a ZPL shift (not shown in the figure). We discuss how this affects the tuning mechanism and provide additional data for several intermediate tuning steps in SI, Sec. 6b.”

.....

Reviewer 2: 10) Many details of the experimental setup are missing or not clearly described, such that it is often not possible to extract what is actually measured, and how. This hinders reproducing the measurements. As an example, it is not mentioned if the HBT measurements use resonant or off-resonant excitation, and if the polarization plot in 3a uses linear or circular polarization in the excitation, or neither of them. Also, which measurements use which filtering setup is not mentioned explicitly. In addition, it is unclear if the data in 3f could be caused by different laser excitation conditions.

Our reply: We thank the Reviewer for this feedback, which has made us realize what further experimental details we were missing. The $g^{(2)}(t)$ measurements were performed pumping above bandgap the on-resonance emitters. The longpass and shortpass filters were installed in the setup to perform all measurements. Additionally, the narrow-band tunable fiber filter was used to perform on-resonance gas-tuned $g^{(2)}(t)$ and lifetime measurements. We now clarify this in the SI, Sec. 2 as follows:

“Additionally, we used a tunable fiber filter from WL Photonics with FWHM transmission bandwidth of 0.10 nm to perform lifetime measurements and gas-tuned second-order correlation measurements on closely cavity-coupled emitters.”

We further stress the point regarding the HBT measurements in the SI, Sec. 5 by stating the following:

“We excited each of our emitters — when closely coupled to their cavities — with a 532 nm CW pump and sent the generated photons to a fiber beam splitter...”

The polarization plot uses a combination of quarter and half wave plates. Also in this case, we now refer in the main text to the SI, Sec. 2, where we comment as follows:

“Both wave plates were used to obtain the polarization plot in Fig. 3a.”

Regarding the data in Fig. 3f, we already explicitly mention that we did not include measurements taken under possibly different laser conditions in our analysis. However, we further stress this point in the SI, Sec. 4 as follows:

“As these measurements were taken under possibly different experimental conditions, we decided not to include them in our theoretical analysis. However, they confirm that the lifetime of our emitter remains essentially unchanged with increasing excitation power. All other reported lifetimes were instead taken under the same laser conditions, which ensures the correctness of our lifetime comparisons.”

We refer the Reviewer to the following point of this response for more details on this last point.

.....

Reviewer 2: 11) On several occasions, the data seems very preliminary, and not taken with the required care. As an example, it is mentioned in the supplement that the measurements were acquired “under likely different laser conditions, originating from an error related to our laser control electronics board.” One would expect that paper data are collected after fixing such errors!

Our reply: Indeed, our analysis was performed using only the data taken under the same laser conditions (with the laser not throwing any error), which ensures the correctness of our lifetime comparison. Therefore, the additional data set that we decided to include in the SI does not play any role in the derivation of our results. This additional set is shown simply for the sake of scientific rigor. We are openly indicating what could have gone differently, and we state this very clearly in the SI, Sec. 4 as follows:

“As these measurements were taken under possibly different experimental conditions, we decided not to include them in our theoretical analysis. However, they confirm that the lifetime of our emitter remains essentially unchanged with increasing excitation power. All other reported lifetimes were instead taken under the same laser conditions, which ensures the correctness of our lifetime comparisons.”

.....

Reviewer 2: Similarly, the data in the main figure are shifted by a random amount, as stated in the supplement: “which resulted in a slight wavelength offset likely due to calibration errors. This offset was taken into account when plotting the data, in order to enable a fair wavelength comparison [...]” One would expect that paper data is taken after a correct calibration of the instruments! Also, what does “a fair comparison” mean here? In agreement with the expectation? How can the authors exclude that the system just degrades/changes over time, which causes deviations from the expectation? Maybe the calibration was correct, but the emitters shifted? This would also question the claimed efficiency change is due to the cavity tuning, rather than to a randomly fluctuating emitter.

Our reply: This comment shows us that this point needs further clarification. As we clearly state in the text, in the temperature tuning case, a grating density of 300 gr/mm was selected, while we used a density of 900 gr/mm when performing gas tuning. The initial calibration of the 900 grating resulted in a small offset compared to the 300 grating, meaning that if we were to measure a known laser wavelength using the 300 gr/mm, say a certain λ_0 , we would always measure $\lambda_0 + c$ with the 900 grating, with c being a constant offset. Therefore, the wavelength reading of the two different gratings has been corrected after calibration against a common reference wavelength. In this sense, this offset is just a systematic error (not random) that we can always correct for, and this is exactly what we did in our experiment. “Fair comparison” does not mean “in agreement with the expectations”, but simply a fair comparison of wavelengths when using different gratings. In this sense, this has absolutely nothing to do with the emitter degrading/shifting over time. If that were the case, we would observe the same shift with both gratings, just offset by a certain constant amount c in one of the two cases. Additionally, the same grating density was used in the same tuning case.

.....

Reviewer 2: 12) In one of the cavities, the authors observe additional Fabry Perot resonances. In the other system, albeit designed identically, this is not observed. What is the reason for this? How can it be excluded that the emitter decays to both sides of the cavity, but the emission is channeled to the detector or away from it by these additional resonances of the setup? It seems that in Fig. 3e, the maximum and minimum emission coincides with maxima and minima of the FP mode. This may be a coincidence, but it is definitely an effect that will obscure the true enhancement of the emission via the resonator. Therefore, the claimed enhancement of the emission is very doubtful to me.

Our reply: The Fabry-Pérot oscillations are not observed in the thermal tuning case because a smaller grating density, and therefore lower resolution, was used. We explain this in the SI, Sec. 6b, stating that the Fabry-Pérot oscillations are “*not observed in the thermal tuning case because of the lower resolution arising from a smaller grating density.*” Moreover, the Reviewer seems to suggest that these additional resonances may have an effect in enhancing or de-enhancing the emission because “*in Fig. 3e, the maximum and minimum emission coincides with maxima and minima of the FP mode.*” However, Fig. 3e shows exactly the opposite. The maximum and minimum emission coincide with a minimum and maximum of the FP mode, respectively. This would mean that, if these additional resonances were to play a role — which is in principle possible — we would be anyway showing the worst case scenario, and not a coincidence where the true enhancement may be obscured by them. We further discuss how these FP resonances may affect the emission in the SI, Sec. 6b, which contains now additional data — corresponding to several tuning steps — showing more cavity profiles and ZPL spectra.

.....

Reviewer 2: 13) On page 2, the authors claim that their $g(2)$ value is “nearly an order of magnitude lower than the rest of the literature”. This is heavily overclaiming and misleading. It ignores a huge amount of the literature, actually all emitters except the G-center, and also Hollenbach et al that is only a factor of 2 off with the same emitter.

Our reply: Since the defect under consideration in our work is the G-center, we are not referring to all the existing literature about the entire field of color centers. In the main text, together with this claim, we immediately refer the reader to the comparison table, which specifies that we are providing a “Comparison of measured properties for the reported G-centers in the literature.”. However, the Reviewer’s comment shows us that this point needs to be further clarified. For this reason, we now state that we are referring to the G-center literature in the main text as follows:

“This value is nearly an order of magnitude lower than the rest of the literature for G-centers (see SI Table I), ...”

Moreover, our claim that we achieve a nearly an order of magnitude lower $g^{(2)}(0)$ is supported by other measured $g^{(2)}(0)$ values reported in Table I. It is clear from the table that we are not ignoring Hollenbach et al. Instead, we are openly providing the value they measured, and pointing out that is however background-corrected, while the other reported values, including ours, are not. This means that their $g^{(2)}(0)$ value with no background correction is surely bigger than 0.07. For this reason, and after this clarification, we believe that claiming that our measurement is *nearly* an order of magnitude lower than the rest of the works on G-centers presented in the table is a fair statement.

.....

Reviewer 2: 14) The authors observe bunching with a 10 ns timescale, which they attribute to a metastable triplet state. This claim is highly doubtful. If it was such triplet state, why would it have a different amplitude in the two studied devices? Why would it have a different timescale, comparing to other works of the literature? How can the authors exclude spectral diffusion, another very common source of bunching in such measurements?

Our reply: We are thankful to the Reviewer for bringing this up. This helps us clarify our $g^{(2)}(t)$ measurements and their interpretation. As we write in the SI, the two $g^{(2)}(t)$ data sets presented in our manuscript are measured on two different devices and at two different powers — 6 μ W (10 μ W) in the thermal (gas) tuning case. Moreover, the emission was narrow-band filtered only in the gas tuning case (to improve the measurement compared to the other case). This makes it already tricky to compare the two measurements, as they were taken under different experimental conditions. For example, as we explain in point 23 of this response, $g^{(2)}(t)$ measurements taken at different powers show different bunching and anti-bunching behaviours, and therefore different amplitudes [Redjem, W. et al. *Nat. Electron.* **3**, 738-743 (2020), Hollenbach, M. et al. *Opt. Express* **28**, 26111-26121 (2020), Wu, E. et al. *Opt. Express* **14**, 1296-1303 (2006), Kurtsiefer, C. et al. *Phys. Rev. Lett.* **85**, 290 (2000)]. Moreover, the \sim 10 ns timescale is in line with what reported in [Hollenbach, M. et al. *Opt. Express* **28**, 26111-26121 (2020)], where they measure \sim 15 ns for single G-centers. More details about the (anti-)bunching behaviour can be found in point 23. In our case, the different powers would explain the different amplitudes between the two measurements, and would therefore not exclude the presence of a metastable state, which other works have taken as a reasonable explanation for the presence of the bunching, see e.g. [Redjem, W. et al. *Nat. Electron.* **3**, 738-743 (2020), Hollenbach, M. et al. *Opt. Express* **28**, 26111-26121 (2020), Komza, L. et al. *arXiv:2211.09305* (2022)] for silicon. However, we agree that the bunching could still be related to spectral diffusion, and we thank the Reviewer for bringing this to our attention. We now clarify this aspect in the SI, Sec. 5 as follows:

*“The G-center is expected to feature three energy levels: a ground and excited singlet state and a metastable triplet state [Udvarhelyi, P. et al. *Phys. Rev. Lett.* **127**, 196402 (2021)]. The metastable state may introduce a bunching effect in the second-order correlation, and has been previously observed for in related works [Redjem, W. et al. *Nat. Electron.* **3**, 738-743 (2020), Hollenbach, M. et al. *Opt. Express* **28**, 26111-26121 (2020)]. This bunching effect is clearly visible in Fig. S7b, while it is less evident in Fig. S7a. We attribute this difference to the fact that the two measurements were taken at different optical powers, and it is therefore not straightforward to make a direct com-*

parison. Moreover, different emitters have different electron trapping rates and other mesoscopic properties that dictate the bunching. Although the presence of a metastable state would explain the observed bunching and is in line with what formulated in previous works [Redjem, W. et al. *Nat. Electron.* **3**, 738-743 (2020), Hollenbach, M. et al. *Opt. Express* **28**, 26111-26121 (2020)], we would like to stress that others factors such as spectral diffusion may play a role in dictating the bunching properties.”

.....

Reviewer 2: 15) It seems that the cavities and emitters cannot be tuned reliably, but it is not mentioned what is causing this, and if this can be overcome. This may hinder reproducing the experimental results even when the same device is used! So, the authors should really explain this in detail.

Our reply: We thank the Reviewer for bringing this up, and we refer to point 9 of this response, which elaborates on this matter in great detail. Interestingly, while the cavity resonance can be tuned reliably, we also observe spectral shifts of the emitter’s ZPL. This phenomenon aligns with what already observed in [Prabhu, M. et al. *Nat. Commun.* **14**, 2380 (2023)]. Consequently, the cavity and emitter may remain closely coupled at certain tuning steps, making it challenging to discern significant variations in ZPL intensity. This situation results in the need for additional tuning steps to achieve the desired (de-)coupling. We delve into this aspect in the SI, Sec. 6b, where we provide additional data illustrating the outcomes of intermediate tuning steps. There, we also mention alternative methods to overcome this challenge.

.....

Reviewer 2: 16) From the Q/V ratio, formula (2), one would expect a Purcell factor of several hundred. Assuming that the dipole moment of the optical transition is reduced 6 fold by the Debye Waller factor, and 5 fold by the extracted quantum efficiency, one would still expect at least a tenfold lifetime reduction. This is in striking contradiction with the constant radiative lifetime! This deviation between theory and experiment is not mentioned in the text, not even in the supplement. To me, it again makes the claim that the observed enhancement of the fluorescence is due to the cavity enhancement very doubtful.

Our reply: We thank the Reviewer for bringing this up, and we are happy to clarify this point in what follows. In Eq. 2 we provide the formula for the Purcell factor in the case of perfect cavity-atom coupling, as already stated in the text. This is the ideal case, which does not take into account the degree of angular and spatial overlap between the dipole moment and the cavity mode electric field. Therefore, even considering the reductions due to the Debye Waller factor and the quantum efficiency, extracting the Purcell factor from Eq. 2 would still lead to an overestimate. On top of this, the Reviewer is considering the value of the quantum efficiency derived in the SI. We highlight that this is not an estimate of the quantum efficiency, but an upper bound. However, even assuming good cavity-atom alignment and highest estimated quantum efficiency, the

observance of no change in lifetime can be explained with the presence of non-radiative decay channels. If their effects are predominant on radiative decay mechanisms, the effectiveness of the expected Purcell enhancement would be reduced. In our experiment, we attribute this mainly to the presence of a metastable state, i.e. level 3 shown in Fig. 1c. When the excited state (level 2) is populated, the emitter can either decay to the ground state (level 1) through radiative processes, that is emitting photons, or to level 3 through non-radiative processes. If the latter effects are large enough, the effective Purcell enhancement would be shadowed by the non-radiative decay rate, and no significant change in the excited state lifetime would be observed. Our observation matches what recently reported in [Lefaucher, B. et al. *Appl. Phys. Lett.* **122**, 061109 (2023)] for ensembles of G-centers in silicon. We now discuss this in the Discussion section of the main text:

*“In our experiments however, no significant change in the excited state lifetime is observed. These results align with what recently reported for ensembles of G-centers [Lefaucher, B. et al. *Appl. Phys. Lett.* **122**, 061109 (2023)]. In general, potential angular and spatial mismatch between the emitter’s dipole moment and the cavity mode electric field would contribute to reduce the Purcell effect. However, we attribute the observance of constant lifetime mainly to the presence of strong non-radiative decay processes, which compete with radiative decay and thus effectively reduce the Purcell enhancement. Such non-radiative processes may be attributed to the presence of a metastable state in the G-center configuration, where the atom can decay to without emitting photons.”*

.....

Reviewer 2: 17) The authors claim that “The quantum efficiency (QE) of any silicon color center is one of the central unanswered questions in the field.” I think that this is not the case – there are many works that investigate this, both for Erbium emitters and other color centers in silicon. However, the values obtained in the literature are often not compatible, which sheds some doubts in the used methods in this and other papers. So the authors should clarify why they think that their method is better than previous measurements, in particular in view of the mentioned weaknesses of their analysis.

Our reply: We thank the Reviewer for this valid point, which has made us realize that limiting our assertion to G-centers would be more fitting. In the case of G-centers, the values reported in the literature are actually not incompatible, as they refer to lower or upper bounds. For example, [Prabhu, M. et al. *Nat. Commun.* **14**, 2380 (2023)] extract a quantum efficiency larger than 0.01, and [Komza, L. et al. *arXiv:2211.09305* (2022)] find a lower bound of 0.02. [Lefaucher, B. et al. *Appl. Phys. Lett.* **122** (2023)] report less than 0.1 for ensembles of G-centers. We are adding to previous works following the same procedure described in [Lefaucher, B. et al. *Appl. Phys. Lett.* **122** (2023)], and thus extracting an upper bound of ~ 0.18 in the case of single G-centers coupled to a cavity. In light of this comment, we have rephrased our statement as follows:

“The quantum efficiency (QE) of G-centers is a central question in the field.”

.....

Reviewer 2: 18) In the discussion, the authors claim that “We show, for the first time, strong enhancement of the quantum emission of individual artificial atoms coupled to silicon nanocavities”. This is again heavily overclaiming! Dibos et al and Ourari et al have measured almost thousandfold lifetime reductions! Redjem and Gritsch have also clearly demonstrated this. The authors mention these works after the manuscript as being published “during the preparation of this manuscript” – however, they were published on the ArXiv several months before the submission, so there would have been ample of time for an adequate discussion, and a comparison of the different works. In particular since the manuscript under consideration does not compete favorably, as it does not show a lifetime reduction that is the smoking gun of Purcell enhancement.

Our reply: We agree with the Reviewer that a more thorough discussion about how our work compares to the rest of the literature, and especially to the ones we became aware of during the preparation of our manuscript, is necessary. In the Introduction, we now make a clear distinction between erbium-based emitters and color centers in silicon. This clarifies how our work differs from [Gritsch, A. et al. *Optica* **10**, 783-789 (2023)]. We then extensively discuss the results of [Redjem, W. et al. *Nat. Commun.* **14**, 3321 (2023)] touching upon the following points:

- i) due to the discrepancy between what they and other works measure (see Table I in the SI, Sec. 8), we suggest the possibility of two different types of G-centers being investigated;
- ii) they do observe lifetime shortening, however their reduced lifetime is ~ 7 ns, which is comparable to our measurement of ~ 6 ns even without lifetime reduction;
- iii) their reported counts are 20 kcounts/s, although details such as the detector efficiency seem to miss. However, even assuming low detection efficiency, this number would still be at least comparable to what we observe (in the case of higher detection efficiency, our system would be notably brighter). They also run the $g^{(2)}(t)$ measurement at 3.3 times the saturation power to get more counts, as they state in the SI. We excited our emitter with only 6 up to 10 μW — well below saturation power — and still recorded very clean data. This suggests a high signal-to-noise ratio in our case.

We therefore highlight that even in the absence of lifetime reduction, we obtain results at least comparable to what [Redjem, W. et al. *Nat. Commun.* **14**, 3321 (2023)] reported. In fact, the single-photon emission reported in our work is notably bright and highly pure. Additionally, we extensively report on the possibility of different types of G-centers and provide an analysis of quantum efficiency. Point i) is extensively addressed throughout our manuscript. We now touch upon point ii) in the SI, Sec. 8 by stating the following:

*“We also point out that Ref. [Redjem, W. et al. *Nat. Commun.* **14**, 3321 (2023)] reports a lifetime shortening from 54 ns down to 7 ns when reducing the coupling between the cavity and emitter. The shortened lifetime value is comparable to our measured values even in the absence of lifetime reduction.”*

Point iii) is already addressed in point 5 of this response.

However, we acknowledge that, if talking about artificial atoms in general, our claim is too strong. We have therefore rephrased it as follows:

“We show intensity enhancement of G-centers coupled to silicon nanocavities and highly pure and efficient single-photon emission.”

.....

Reviewer 2: 19) The authors claim in the outlook that “Our demonstration lays the groundwork for efficient spin-photon interfaces” – this is very bold, given that no efficiency is reported (and in the supplement one finds it is probably very lousy), and in addition the studied emitter does not even have a spin! In particular in view of other works that have clearly observed single spins in silicon with high efficiency (Higginbottom 2022, Gritsch 2023), it is completely unclear to me how the authors could arrive at such claims.

Our reply: In light of several previous points raised by the Reviewer, which led us to modify a substantial part of the Introduction, we have now changed parts of the Conclusion as well to make it consistent with the new Introduction. We now make it more clear that our system is an efficient source of single photons and does not possess an addressable spin, although we suggest possible future directions where the anticipated spin in the metastable state may be investigated, and that our demonstration could be directly applied to other color centers which possess a spin, e.g. the T centers. Our Conclusion now reads as follows:

*“We showed cavity-enhanced single artificial atoms in silicon by integrating single G-centers into inverse-designed photonic crystal nanocavities. We demonstrated a 6-fold intensity enhancement as well as the highest purity single-photon source for G-centers in the literature, and the first bound to the QE of single cavity-coupled G-centers of < 18%. New directions may also involve the investigation of the anticipated spin in the metastable state to realize G-center based spin-photon interfaces. However, our device could already be suitable to implement spin-photon interfaces based on e.g. T-centers, which possess an addressable spin in the ground state [Higginbottom, D. B. et al. Nature **607**, 266–270 (2022)]. Moreover, we shed light on new properties of single artificial atoms in silicon, suggesting the possibility of two different types of G-center-like structures observed so far. Despite G-centers — and artificial atoms in silicon in general — being still under investigation, the impressive amount of literature generated in the past few years suggests that emitters in silicon may indeed hold promise as viable candidates for practical integration and large-scale quantum information processing [Osika, E. N. et al. Phys. Rev. Appl. **17**, 054007 (2022), Bogaerts, W. et al. Nature **586**, 207–216 (2020), Silverstone, J. W. et al. Nat. Photon. **8**, 104–108 (2014)].”*

We thank the Reviewer for the detailed feedback on this point, which has led, in our opinion, to a substantial improvement of the Conclusion section as well.

.....

Reviewer 2: 20) In the final part of the outlook the authors mention a collection of potential applications of their system. However, I cannot make sense of any of the mentioned topics, how the current work would relate to them! Ref 29 and 30 deal

with photonic circuits that are programmable by temperature or electro-optic effects – I cannot see how the current work would add to this! Also, the mentioned photonic quantum information processing devices are clearly out of reach if the emitter is not coherent, which, following Komza et al, is very likely. Ref 34 is not even about deterministic entanglement generation, but about photonic cluster states. Ref 33 is about self-learning algorithms – how would a G-center come in there? So, to summarize, the whole last paragraph does not seem to make any sense, and rather leaves the impression of a random collection of buzzwords.

Our reply: We thank the Reviewer for bringing this to our attention. In the final part of our manuscript, we were not referring to G-centers specifically, but rather to the field of silicon artificial atoms in general. The references about programmable photonic circuits were cited to highlight the integration potential of silicon artificial atoms in complex programmable circuits. When discussing quantum information processing, we were again not only referring to G-centers, whose properties are still under investigation. For example, it is still unclear whether a predicted spin in the metastable state can be used for applications. However, the huge amount of effort that numerous research teams are nowadays devoting to this field suggests that emitters in silicon may indeed hold promise as viable candidates for large-scale quantum information processing. In light of this and other comments, we have substantially changed the Conclusion section and therefore removed some of the references (see previous point of this response).

.....

Reviewer 2: 21) The authors only mention in the methods that the fabrication of G-centers seems to unavoidably reduce the Q-factor of their nanophotonic devices. This leads to an inherently low efficiency. Why is this not mentioned and discussed in the main text, in particular since the whole paper is about the claim of enhancing efficiencies?

Our reply: We agree with the Reviewer that this information should be included in the main text as well. We now discuss this also in the main text as follows:

*“The measured quality factors are much lower than $\mathcal{O}(10^6)$, which is what expected from simulations and already measured similar samples [Panuski, C.L. et al. Nat. Photon. **16**, 834-842 (2022)]. We attribute this to the damage induced by the carbon implantation process. A more detailed discussion about this, as well as suggested approaches to improve or avoid this problem, can be found in Methods, Sec. B.”*

.....

Reviewer 2: 22) The authors claim that they achieve record-low $g(2)$, but they only do this on a single devices! Already the second one they study performs much worse. This should also be mentioned in the main text, instead of cherry-picking the best and then claiming any records.

Our reply: In the gas tuning case, the low value of 0.03 comes from an improvement of our measurement apparatus compared to the initially studied temperature tuning case. The $g^{(2)}(t)$ measurements performed in the temperature tuning case are indeed taken without narrow-band filtering, which explains the worse performance. In this sense, the second measurement we take actually performs better than the first one, and not the other way around. In the main text, we already refer the reader to the SI, Sec. 5 for more details on that particular measurement. However, the Reviewer’s comment shows us that this point needs to be further clarified. Therefore, we now specify this also in the main text by stating the following:

“Our second-order autocorrelation results (Fig. 3c) show excellent antibunching with a fitted $g^{(2)}(0)$ value of $0.03_{-0.03}^{+0.07}$ without background correction (this number refers to the gas tuning case and results lower than the $g^{(2)}(0)$ measured in the temperature tuning case thanks to a setup improvement, see details in SI, Sec. 5).”

.....

Reviewer 2: 23) In the supplement, they extract the lifetime from the $g^{(2)}$ measurement. It is only 3.16 ns, so far from being consistent with the pulsed measurements (6 ns in 3f, 4.8ns in S3). This contradiction has to be resolved, in particular since the lifetime is the central element that goes into the QE analysis!

Our reply: We are thankful to the Reviewer for raising this point. We understand that this aspect requires additional clarification and we are happy to address this both here and in the manuscript. In quantum photonics experiments, second-order correlation measurements are the gold standard to reveal the quantum nature of the emission, and it is in general not common to extract lifetime information from them. As an example, we can consider performing $g^{(2)}(t)$ measurements at different powers. This may result in a change of the emitter’s anti-bunching behaviour, namely the width of the anti-bunching dip. As a consequence, this would impact the apparent excited state lifetime τ_1 extracted from the $g^{(2)}(t)$ fits. It is only when the optical power tends to zero that the actual lifetime of the excited state can be extracted, see [Redjem, W. et al. *Nat. Electron.* **3**, 738-743 (2020), Hollenbach, M. et al. *Opt. Express* **28**, 26111-26121 (2020), Wu, E. et al. *Opt. Express* **14**, 1296-1303 (2006), Kurtsiefer, C. et al. *Phys. Rev. Lett.* **85**, 290 (2000)]. Hence, it would be necessary to obtain the lifetime from $g^{(2)}(t)$ measurements at different powers, and then extrapolate the value at zero optical power. In more detail, the decay rate $1/\tau_1$ would decrease with the optical power. Therefore, the actual lifetime extracted from the $g^{(2)}(t)$ data would be higher than the fitted value of 3.16 ns, in accordance with what we expect. For this reason, the lifetime value we extract from the $g^{(2)}(t)$ fit is not representative of the actual excited state lifetime. For that purpose we perform pulsed laser measurements instead, which are commonly used to reliably extract lifetime values. We now make this clear in the SI, Sec. 5 by stating the following:

“We stress here that $g^{(2)}(t)$ measurements at different powers and subsequent extrapolation of the decay rate from the excited to the ground state would be needed to obtain reliable lifetime values [Redjem, W. et al. *Nat. Electron.* **3**, 738-743 (2020), Hollenbach, M. et al. *Opt. Express* **28**, 26111-26121 (2020), Beveratos, A. et al. *In Quantum Communication, Computing, and Measurement* **3**, 261–267 (Springer, 2002)]. Hence,

the lifetimes extracted from this fit are not to be considered an indication of the actual values. We only perform $g^{(2)}(t)$ measurements to obtain information about the nature of the emission. To extract reliable lifetime values, we perform pulsed laser measurements instead.

.....

Reviewer 2: In addition to these main criticisms, I also have several smaller remarks that the authors should consider before resubmitting: 1) Fig 1 suggests that the authors have many identical devices on the same chip, but the actual measurements are performed only on two devices, so Fig. 1 is misleading.

Our reply: We thank the Reviewer for this remark. We would like to point out that Fig. 1 is indeed a true representation of our work, as it shows one chip hosting many cavities (as discussed in the SI, Sec. 1 and Sec. 2) with two of them being investigated with laser beams, which indeed represents what was done in our study.

.....

Reviewer 2: 2) I suggest to plot the spectrum in Fig. 1 on a log scale, such that the phonon sideband is visible. Otherwise, the reader gets the wrong impression that the Debye-Waller factor is close to unity

Our reply: Following the Reviewer’s suggestion, Fig. 1 contains now a spectrum plot on a log scale.

.....

Reviewer 2: 3) When the authors mention the first observation of spins in silicon, they should also include Chen et al. 2020 and Gritsch et al. 2023 that have measured single spins coupled to nanophotonic cavities.

Our reply: In accordance with the Reviewer’s suggestion, we have now included Chen et al. 2020 and Gritsch et al. 2023 as well.

.....

Reviewer 2: 4) The upper bound to the QE would urgently need an analysis of the statistical and systematic errors.

Our reply: Following the Reviewer’s suggestion, we have now included an error estimate of the upper bound to the QE taking into account Poissonian statistics of the measured counts. This results in an error of $\sim 1\%$ on the bound. We have added the following to the SI, Sec. 7:

“... we can estimate the QE to be bounded by $(18\pm 1)\%$ for a measured off-resonance lifetime value of $\tau_{off} = (6.09 \pm 0.25)$ ns. The uncertainty on the bound is derived from propagated Poissonian statistics of the measured counts.”

.....

Reviewer 2: 5) What is the yield of the G-center production mentioned in the methods?

Our reply: In order to make statistically sound conclusions about distributions and yields, large datasets would be critical. We are working to extend our automated spectroscopy techniques [Sutula, M. et al. *Nature Materials* 1-7 (2023)] to the IR regime, but face challenges from IR detection and lower signals as explained below. While preliminary, these studies provide important insights despite limitations on statistical claims. The challenges for automated big-data spectroscopy for us presently are as follows, which makes estimating the yield challenging. First of all, our chip hosts thousands of cavities which differ in size and quality factor, and feature resonance wavelengths spanning a relatively wide range (from ~ 1270 to ~ 1320 nm). Our fabrication approach involves individually modeling each of our cavities through an inverse design process. This means that we do not replicate identical cavity designs; rather, each cavity will differ from all others. The difference in cavity parameters results in most emitters being decoupled, with some cavities featuring potentially too low Q to show enhancement or far off resonance wavelength, or with emitters not being spatially aligned to cavities. Moreover, with full decoupling, our setup cannot detect ZPL emission, obscuring yield estimates.

In this study, we targeted 10 cavities based on the Q factor, size, and resonance wavelength, selecting two that showed single emitters. Data sets with no or multiple emitters were discarded. We already discuss this in the SI, Sec. 2, together with approaches to improve yield through localized implantation, strain tuning, and local thermal oxidation in the main text Discussion. Further work is needed to realize large-scale, deterministic cavity-emitter coupling, which remains an open challenge in the field.

.....

Reviewer 2: 6) Ref [6] gives a reference to the Purcell effect that seems of. The paper they want to cite was published in 1946 rather than 1995. Instead, I would suggest to pick a newer reference to a review paper that covers the Purcell effect, but also its realization with nanophotonic structures, e.g. Lodahl et al *Rev.Mod.Phys.* 2015

Our reply: We agree with the Reviewer that including a newer reference would be beneficial. We have now corrected the original Purcell reference, which we decided to

leave because it refers to the first ever discussed Purcell enhancement, and added [Lodahl, P. et al. *Rev. Mod. Phys.* **87**, 347 (2015)] for the sake of completeness.

.....

Reviewer 2: 7) The authors mention in the supplement that the sample is mounted to a nanopositioning stage. It is a common issue to ensure thermalization in such setting in vacuum cryostats. So, the authors should mention how this is achieved, and where the temperature is measured in their experiments. If it is only the stage, the thermal tuning may have a completely wrong temperature range.

Our reply: We thank the Reviewer for this remark, which gives us the possibility to clarify their concern. Mounting samples onto nanopositioning stages in vacuum cryostats is a standard procedure in many solid-state experiments. These stages exhibit limited thermal conductivity, and therefore it is common to use thermal links between the cold finger (placed beneath the stages) and the sample (mounted on top of the stages) to ensure proper thermalization of the sample. For this reason, the nanopositioning stages do not affect the measured temperature. Samples are usually placed in cold platforms — where temperature sensors are attached. The temperature is thus not measured on the stage, and the sample platform temperature is very stable over time. We have now clarified where the temperature is read (main text) and commented on the thermal links (SI, Sec. 2) as follows:

“In the thermal tuning experiments, starting with a sample platform temperature of 4 K in our cryostat, we brought the temperature up to 24 K...”

“The sample is mounted on a XYZ cryogenic piezoelectric stages (Attocube). As these stages exhibit limited thermal conductivity, thermal links between the cold finger (placed beneath the stage) and the sample (mounted on top of the stage) are used to ensure proper thermalization of the sample.”

In response to Reviewer 3:

Reviewer 3: The manuscript by Saggio et al. reports on the fabrication, characterization, and modeling of G-centers in silicon. Defect centers in silicon are attracting increasing attention as they combine the benefits of silicon photonics and silicon nanotechnology with localized quantum emitters, and the present manuscript reports on valuable additions to this nascent field. The main result is the realization of cavity-enhanced emission including the demonstration of a rather pure single-photon emission. The manuscript is generally clearly written and the figures are mostly readable and clear, but I would need clarification of a number of points before I can assess whether or not it is suitable for publication in Nature Communications.

Our reply: We are happy to read that the Reviewer appreciates the clarity of our work. Below we provide a detailed point-by-point response to address their concerns.

.....

Reviewer 3: These points are – in order of importance: 1. The authors write under Notes that they have become aware of two recent papers on similar defects in cavities. This is not a proper way of discussing highly relevant and related works (one of which has since appeared as a journal publication: <https://www.nature.com/articles/s41467-023-38559-6>). Rather, the authors should discuss these works in the introduction and explain exactly at which points the present work advances the state of the art relative to these works. If the advances relative to previous works are not significant and clear, Nature Communications is hardly the right outlet.

Our reply: We agree with the Reviewer that a more thorough discussion and comparison with other recent works are necessary. We are happy to address the differences in what follows:

- [Gritsch, A. et al. *Optica* **10**, 783-789 (2023)] show cavity coupling of an erbium-based emitter, which is intrinsically different from color centers in silicon (our case). In short, several works such as [Dibos, A.M. et al. *Phys. Rev. Lett.* **120**, 243601 (2018), Ourari, S. et al. *arXiv:2301.03564* (2023), Gritsch, A. et al. *Optica* **10**, 783-789 (2023), Berkman, I.R. et al. *Phys. Rev. Appl.* **19**, 014037 (2023)] have shown that these emitters emit directly into the telecom C-band. However, the lifetime of such systems is very long (in the order of \sim ms in the bulk) and Purcell enhancement has been demonstrated that reduces it down to $\sim \mu$ s (which is for example the case of [Gritsch, A. et al. *Optica* **10**, 783-789 (2023)]). This is still three orders of magnitude higher than what we observe for G-centers, despite the fabrication of cavities with very large quality factors. The intrinsically long lifetime unavoidably leads to low photon emission rates. As discussed in [Dibos, A.M. et al. *Phys. Rev. Lett.* **120**, 243601 (2018)], enormous Purcell factors ($> 10^5$) would be required to enhance the photon emission to 10 MHz, since the initial rate is only in the order of tens of Hz. Similarly, in [Berkman, I.R. et al. *Phys. Rev. Appl.* **19**, 014037 (2023)] they estimate that a Purcell factor of approximately 2×10^6 would

be needed to achieve emission rates of 400 MHz. However, such large Purcell factors have not been demonstrated yet, despite the use of cavities with very high quality factors. We now properly discuss this in the Introduction, making the distinction between erbium-based emitters and color centers clear.

- [Redjem, W. et al. *Nat. Commun.* **14**, 3321 (2023)] focus instead on the Purcell enhancement of a system supposedly similar to ours. We mention this work in the Introduction and then extensively discuss their results throughout our manuscript touching upon the following points:
 - (i) due to the discrepancy between what they and other works measure (see Table I in the SI, Sec. 8), we suggest the possibility of two different types of G-centers being investigated;
 - (ii) they do observe lifetime shortening, however their reduced lifetime is ~ 7 ns, which is comparable to our measurement of ~ 6 ns even without enhancement;
 - (iii) their reported counts are 20 kcounts/s, although details such as the detector efficiency seem to miss. However, even assuming low detection efficiency, this number would still be at least comparable to what we observe (in the case of higher detection efficiency, our system would be notably brighter). They also run the $g^{(2)}(t)$ measurement at 3.3 times the saturation power to get more counts, as they state in the SI. We excited our emitter with only 6 up to 10 μW — well below saturation power — and still recorded very clean data. This suggests a high signal-to-noise ratio in our case.

We therefore highlight that even in the absence of lifetime reduction, we obtain results at least comparable to what [Redjem, W. et al. *Nat. Commun.* **14**, 3321 (2023)] reported. In fact, the single-photon emission reported in our work is notably bright and highly pure. Additionally, we extensively report on the possibility of different types of G-centers, provide an analysis of quantum efficiency and an exhaustive comparison of the recent literature for G-centers. Point (i) is extensively addressed throughout our manuscript. We now touch upon point (ii) in the SI, Sec. 8 by stating the following:

*“We also point out that Ref. [Redjem, W. et al. *Nat. Commun.* **14**, 3321 (2023)] reports a lifetime shortening from 54 ns down to 7 ns when reducing the coupling between the cavity and emitter. The shortened lifetime value is comparable to our measured values even in the absence of lifetime reduction.”*

Point (iii) is commented in the same section as follows:

“Even though a direct comparison of the efficiencies among these works is challenging due to different experimental parameters such as N.A. of the objective, filtering apparatus, and coupling and detection efficiency, we can qualitatively correct for the detection efficiency, when reported, and extract an indicative value of the count rates at saturation. Most of the cited works reporting saturation measurements on single G-centers would measure rates well below ~ 100 kcounts/s. In our case, we would measure ~ 200 kcounts/s instead. While this is just a qualitative estimate, we can conclude that our ZPL enhancement leads to notably bright single-photon emission.”

In summary, these two works are now properly acknowledged in our manuscript, and the differences between their results and ours clearly stated.

.....

Reviewer 3: 2. In Eq. (1) as well as in the model for the quantum efficiency in the SI, the authors seem to define the Purcell-enhanced rate as $(1 + F_P) * \gamma_R$, where F_P is the Purcell factor and γ_R is the radiative rate of the transition. This definition is at least sloppy and perhaps also wrong: The Purcell-enhanced rate is $F_P * \gamma_R$, where γ_R is the radiative rate of an emitter in an infinitely extended medium with the same refractive index as the host material (in this case silicon). I do not understand the factor of one in this expression here and elsewhere in the manuscript. I would ask the authors to explain this model and also to be more rigorous regarding the difference between the radiative rate in the experiment and that in an infinitely extended medium.

Our reply: We thank the Reviewer for raising this point, which we are happy to clarify. The Reviewer is indeed correct that the radiative decay rate γ_R of an emitter placed in a resonant structure is $\gamma_R = F_P \gamma_R^{\text{hom}}$, where γ_R^{hom} is the radiative decay rate of an identical emitter placed in a homogeneous medium with the same refractive index as the host material and F_P is the Purcell factor. Therefore, one has that $F_P = \frac{\gamma_R}{\gamma_R^{\text{hom}}}$. This is a generalization of the originally formulated Purcell factor [Purcell, E. *Phys. Rev.* **69**, 681 (1946)] to the case of arbitrary photonic nanostructures, meaning that F_P includes contributions from the cavity mode emission as well as from the emission from modes outside the cavity [Lodahl, P. et al. *Rev. Mod. Phys.* **87**, 347 (2015), Faraon, A. et al. *Nat. Photon.* **5**, 301-305 (2011)]. Hence, it can be written as $F_P = F_P^{\text{cav}} + F_P^{\text{leak}}$, with $F_P^{\text{cav}} = \frac{\gamma_R^{\text{cav}}}{\gamma_R^{\text{hom}}}$ and $F_P^{\text{leak}} = \frac{\gamma_R^{\text{leak}}}{\gamma_R^{\text{hom}}}$, where γ_R^{cav} is the decay rate into the cavity mode and γ_R^{leak} the decay rate into all other non guided modes. In typical nanocavity experiments, it is common to assume that the emission rate outside the cavity mode matches that in a homogeneous medium, and thus $\gamma_R^{\text{leak}} \approx \gamma_R^{\text{hom}}$ [Faraon, A. et al. *Nat. Photon.* **5**, 301-305 (2011), Crook, A. L. et al. *Nano letters* **20**, 3427-3434 (2020), Li, L. et al. *Nat. Commun.* **6**, 6173 (2015), Riedrich-Möller, J. et al. *Appl. Phys. Lett.* **106**, (2015), Gong, Y. et al. *Opt. Express* **18**, 2601-2612 (2010)]. This leads to rewriting the Purcell factor as $F_P \approx F_P^{\text{cav}} + 1$. Usually, other works may refer to either the generalized Purcell factor F_P , or assume that $F_P^{\text{cav}} \gg F_P^{\text{leak}}$, which explains the absence of the factor +1.

In our manuscript, the F_P we refer to corresponds to the F_P^{cav} mentioned here. We acknowledge that we should be more rigorous about this aspect, and we now address this both in the main text and SI, Sec. 7 as follows:

“In more detail, the modified local density of optical states in a cavity can increase the radiative emission fraction β into a desired mode while suppressing emission into other modes:

$$\beta = \frac{(1 + F_P)\gamma_R}{(1 + F_P)\gamma_R + \gamma_0}, \quad (1)$$

*where γ_R is the radiative rate of an emitter placed in a homogeneous medium with the same refractive index as the host material, and γ_0 encompasses the rates for non-radiative transitions. F_P is the cavity Purcell factor, which in the case of perfect cavity-atom coupling is defined as [Purcell, E. *Phys. Rev.* **69**, 681 (1946), Lodahl, P. et al. *Rev. Mod. Phys.* **87**, 347 (2015)]*

$$F_P = \frac{3\lambda^3 Q}{4\pi^2 V}. \quad (2)$$

As Eq. 1 shows, γ_R is enhanced by a factor $(1 + F_P)$ when the emitter is placed in a resonant structure. This comes from considering both the emission into the cavity mode as well as into all other non guided modes (see SI, Sec. 7 for more details)."

"As already mentioned in the main text, γ_R is enhanced by a factor $(1 + F_P) = F_P^*$ when the emitter is placed in a cavity. This comes from considering F_P^* as a generalization of the originally formulated Purcell factor to the case of arbitrary photonic nanostructures [Lodahl, P. et al. *Rev. Mod. Phys.* **87**, 347 (2015)]. In this case, F_P^* includes contributions from the cavity mode emission as well as from the emission from modes outside the cavity [Faraon, A. et al. *Nat. Photon.* **5**, 301-305 (2011), Lodahl, P. et al. *Rev. Mod. Phys.* **87**, 347 (2015)]. This means that it can be written as $F_P^* = F_P + F_P^{\text{leak}}$, with $F_P = \frac{\gamma_R^{\text{cav}}}{\gamma_R}$ and $F_P^{\text{leak}} = \frac{\gamma_R^{\text{leak}}}{\gamma_R}$, where γ_R^{cav} is the decay rate into the cavity mode and γ_R^{leak} the decay rate into all other non guided modes. In typical nanocavity experiments, it is common to assume that the emission rate outside of the cavity mode matches that in a homogeneous medium, and thus $\gamma_R^{\text{leak}} \approx \gamma_R$ [Crook, A. L. et al. *Nano letters* **20**, 3427-3434 (2020), Li, L. et al. *Nat. Commun.* **6**, 6173 (2015)]. This leads to rewriting the generalized Purcell factor as $(1 + F_P)$."

Reviewer 3: 3. For the cavity optimization, the authors refer to their previous work (Ref. 17) but the present manuscript needs to include more information on this as the present discussion is limited to Fig. S1. In particular, the authors should explain what is meant by the "0th Brillouin zone."

Our reply: We thank the Reviewer for bringing this to our attention. We agree that more information on our cavity optimization process needs to be included, as the process enables a new class of optical cavities that simultaneously possess high-Q and vertical beaming. We have now extensively expanded Sec. 1 of the SI, adding more figures and commenting more on the cavity optimization procedure. As the section has been substantially extended, we refer the Reviewer to the SI, Sec. 1 for details. There, we have updated the "0th Brillouin zone" terminology to "zero-order diffraction efficiency" in line with the defined optimization objective function and process.

Reviewer 3: 4. It can be tricky to make proper fits to decay curves, but only a single decay curve is shown in the SI, and this is insufficient. The main text should include such raw data including the fit, both for the case of on- and off-resonance emission.

Our reply: We understand the need for showing additional lifetime fits, and we have now added all data and fits used in the analysis in the SI, Sec. 4.

Reviewer 3: 5. On a related note: Figure 3f is hard to read because the horizontal ticks are in deltas while the axis label advertises this as being in nanometers.

Our reply: We thank the Reviewer for raising this point, which we are happy to address. The x-axis labels are correct because δ is a detuning, and is therefore expressed in nanometers. Each tick corresponds to a different detuning. However, we acknowledge that Fig. 3f might be a little hard to read due to the amount of information it contains. We have therefore expanded on this in the figure caption as follows:

“PL counts and lifetimes for both systems (thermally and gas tuned) under the measured cavity-atom detuning δ (nm). δ_t and δ_g indicate the detunings for the thermal and gas case, respectively, and the prime is used to differentiate between the two different detunings within the same case.”

.....

Reviewer 3: 6. Figure 2 shows a number of figures including data on color scales – but without showing the color scale. This becomes particularly troublesome for Fig. 2d, which includes two unidentified color scales on top of each other. Surely, this can be plotted in a better way while showing all color scales.

Our reply: We thank the Reviewer for this comment, which has helped us improve the overall clarity of Fig. 2. We have now updated the previous subfigures and added appropriate color scales.

.....

Reviewer 3: 7. The authors should include information about how many samples were characterized, if any data sets were discarded, etc. The SI discusses these important points in too vague terms.

Our reply: We agree with the Reviewer that this point needs further clarification. Our sample hosts thousands of cavities, fabricated in different sizes, spanning a relatively wide resonance wavelength range (from ~ 1270 to ~ 1320 nm), and optimized for different parameters such as the Q-factor and the far-field profile. Our fabrication approach involves individually modeling each of our cavities through an inverse design process. This means that we do not replicate identical cavity designs; rather, each cavity will differ from the others. In our study, only one sample was used and a few cavities (~ 10) were targeted based on the Q-factor, size and resonance wavelength. Each cavity was then probed to look for the presence of a single emitter. Due to the intrinsically random carbon implantation process, not every cavity will feature the presence of a single emitter (we already discuss methods to improve this in the main text Discussion.) In our work, we discarded data sets that did not show the presence of an emitter, or that showed multiple ones. As the Reviewer suggests, we now discuss this point in the SI, Sec. 2:

“In our experiment, one sample hosting thousands of cavities — each nominally different from all others — was used. After targeting a few cavities (~ 10) based on Q -factor, size and resonance wavelength, each of them was probed to look for the presence of a single emitter. Two of them were then chosen and analyzed. We discarded data sets that did not show the presence of an emitter, or that revealed multiple ones.”

.....

Reviewer 3: 8. The reported quantum efficiencies in a number of related works are stated in the final table, but it appears not to be representative. In particular, Ref. 28 states that “The lifetime reduction and Purcell acceleration observed in our work for a single center indicates a close to unity quantum efficiency.” While it is clear that this quantitative statement is hard to represent in a table, I am not convinced that leaving out this number when listing Ref. 28 is a fair representation of Ref. 28. If there are issues with the model used in Ref. 28 or if there is some other reasoning behind ignoring this statement, it should be explained in section 8 of the SI.

Our reply: The reason as to why this number was not included in the table is simply that the authors of Ref. 28 did not include this statement in their arXiv version, which was the only available version of their work at the time of our submission to *Nature Communications*. They added this estimate of quantum efficiency after the peer review process, and we therefore were not aware of this number when we first submitted. We thank the Reviewer for pointing this out, and we have now included this number in the table to ensure fair comparison.

REVIEWER COMMENTS

Reviewer #1 (Remarks to the Author):

Title: Cavity-enhanced single artificial atoms in silicon

Authors: Valeria Saggio, Carlos Errando-Herranz, Samuel Gyger, Christopher Panuski, Mihika Prabhu, Lorenzo De Santis, Ian Christen, Dalia Ornelas-Huerta, Hamza Raniwala, Connor Gerlach, Marco Colangelo, and Dirk Englund

In the revised version of this manuscript, the authors have made a thorough and largely convincing response to my original questions. Additionally, they have added a significant amount to the manuscript text for additional clarity. The revised version is much stronger than the original submission, and I now believe it is close to being ready for acceptance. However, I do still have a few outstanding questions that should be addressed before I am ready to accept the manuscript for publication.

1. In response to my original comment on the discrepancy between the count rate measured in the saturation curve in Fig. 3b and the counts vs. detuning in Fig 3f., the authors explained that these data were collected using different spectrometer gratings. However, measuring count rates using a spectrometer is often less reliable than with a single photon counter as doing so requires careful calibration. The authors do have SNSPDs to measure photon statistics with: could they provide more insight on why they chose to measure count rates with their spectrometer and how it was calibrated?

2. The authors argue that the dominance of non-radiative decay channels is responsible for the constant lifetime of the G-centers, even when tuned into resonance with their cavities. This is certainly a possibility, although it is common that both radiative and non-radiative decay paths contribute to excited state lifetime measurements – which can be seen as a bi-exponential decay curve. The authors provide excited state lifetime measurements in the supplementary information, but they fit using a single exponential distribution. Have the authors tried fitting lifetime measurements from the emitter on and off resonance with a bi-exponential? Moreover, have they measured their setup instrument response function and used any deconvolution algorithms to extract a more accurate lifetime fit? Doing so could potentially illustrate the change in the decay time of the radiative transition.

As I have mentioned earlier, I think this research is a big step forward in the world of quantum technology using silicon-based artificial atoms. I'm looking forward to seeing the updated manuscript and how my questions have been addressed.

Reviewer #2 (Remarks to the Author):

The authors have addressed many (but not all) of my criticisms. In particular, they have reduced the claims and put the work into the right perspective. However, with this reduction – albeit appropriate – I'm afraid that Nature Communications is not a suited journal, as it seems that the work is not relevant for the majority of the community. To summarize, it does not add anything to the literature concerning emitters in the solid. The findings even fail to be of relevance for the subfield working on emitters in silicon... not even for the groups working in color centers. Instead, it seems that the results are only of interest for the few groups that work on G-centers, or – even more narrow – on a sub-class of this specific defect. This narrow scope is in contradiction with the targeted scope of the journal, such that I can only recommend to resubmit the work to another journal that is more focused.

Reviewer #3 (Remarks to the Author):

The authors have carefully addressed most of my comments, and I appreciate the care and seriousness put into the response. I also find the manuscript greatly improved. However, I maintain some criticism on two points. The first may be conceptually important but perhaps not so significant for the results. The second point is more important because the claims of the paper do not match the results. The paper contains important high-quality data that clearly deserve publication, but I cannot see that Nature Communications is a proper outlet unless the authors significantly revise their claims to be consistent with their experimental findings: In contrast to their claims, their data shows that G centers are not very promising for quantum technologies and that extreme Purcell factors are needed to bring them into a regime of even remotely useful quantum efficiencies.

The response provided by the authors on the discussion of the Purcell factor, $(1 + FP) * \gamma_R$, is not correct. I now understand where it comes from, and this is approximately valid for many types of microcavities and macroscopic cavities where the spectrum of the LDOS is approximately a peak on a background. In this case, the background can be approximated by a uniform medium, which after normalization enters (in the Purcell factor) as an additional term of unity. But the authors consider a photonic crystal cavity where this approximation does not hold as the background LDOS is strongly suppressed by the (in-plane) photonic band gap, which reduces the background LDOS by 10-100x dependent upon position and polarization (and to a minor degree also on wavelength). The authors should address how this impacts their findings.

My second point regards the claims of the paper in relation to the quantum efficiency. I am fully aware that it has become customary to make inflated claims about paving the way for some kind of quantum technologies in nearly all papers dealing with quantum systems, but it does not make it meaningful. The implication of the author's observations is that while the g^2 looks impressive and useful, the decay is predominantly non-radiative, which is why they do not observe any enhancement of the measured decay rate. Since the measured lifetime is on the order of a few nanoseconds, the non-radiative decay must be much faster (as also discussed explicitly by the authors). In their discussion of erbium emitters, they mention their very long lifetime as a downside, and I agree, but it actually implies an extremely small non-radiative decay rate. So it appears that erbium emitters must be greatly enhanced to increase the radiative rate and the color centers studied by the authors must be greatly enhanced to combat the large non-radiative rate. In other words, there is no obvious advantage of the G centers in this regard. Now, the paper is not about erbium so this is less important, but, on this background, a statement like "Our results show intensity enhancement of G-centers as well as highly pure and efficient single-photon emission, paving the way towards scalable quantum information processing" seems to be simply incorrect. Their results indicate on the contrary that G-centers are not very promising (modulo the possibility of the existence of several kinds of G centers) for quantum technologies. A similar comment applies to the claim of "cavity enhancement" of single-photon emission, because the data shows only reshaping of the far field through their nicely designed cavity, but no Purcell effect.

Response to the Referee reports for manuscript NCOMMS-23-20767A.

We thank the Reviewers for the time they took to evaluate our revised manuscript. Below we address all of their comments in a detailed point-by-point response.

.....

In response to Reviewer 1:

Reviewer 1: In the revised version of this manuscript, the authors have made a thorough and largely convincing response to my original questions. Additionally, they have added a significant amount to the manuscript text for additional clarity. The revised version is much stronger than the original submission, and I now believe it is close to being ready for acceptance. However, I do still have a few outstanding questions that should be addressed before I am ready to accept the manuscript for publication.

Our reply: We are happy to read that the Reviewer is satisfied with the work we have done to address their questions and concerns. We provide detailed responses to the additional comments below.

.....

Reviewer 1: 1. In response to my original comment on the discrepancy between the count rate measured in the saturation curve in Fig. 3b and the counts vs. detuning in Fig 3f., the authors explained that these data were collected using different spectrometer gratings. However, measuring count rates using a spectrometer is often less reliable than with a single photon counter as doing so requires careful calibration. The authors do have SNSPDs to measure photon statistics with: could they provide more insight on why they chose to measure count rates with their spectrometer and how it was calibrated?

Our reply: We agree with what is stated by the Reviewer, and we are happy to further clarify this point. The reason as to why we measured our count rates with the spectrometer instead of using SNSPDs was related to temporal unavailability of our single-photon detectors due to maintenance. However, performing this type of measurements with the spectrometer leads to accurate results as well, especially because we are not interested in the absolute count rate values, but only in the relative difference between them. To carry out such measurements, the spectrometer was calibrated against a reference lamp shone into its aperture. We then used a fully-automated calibration system (Intelli-Cal) integrated in the spectrometer software, which allows for accurate both wavelength and intensity calibration. This method has the advantage of eliminating the instrument response, leaving only the sample response. Moreover, as opposed to traditional interpolation methods that use just a few peaks for calibration, this feature calibrates the entire spectrum, and thus provides 10 times greater accuracy. In this way, reliable intensity and wavelength calibration is achieved.

.....

Reviewer 1: 2. The authors argue that the dominance of non-radiative decay channels is responsible for the constant lifetime of the G-centers, even when tuned into resonance with their cavities. This is certainly a possibility, although it is common that both radiative and non-radiative decay paths contribute to excited state lifetime measurements – which can be seen as a bi-exponential decay curve. The authors provide excited state lifetime measurements in the supplementary information, but they fit using a single exponential distribution. Have the authors tried fitting lifetime measurements from the emitter on and off resonance with a bi-exponential?

Our reply: We thank the Reviewer for bringing this up and giving us the opportunity to clarify this important point. The Reviewer is commenting about the possibility of fitting the lifetimes to bi-exponential decay curves, instead of mono-exponential functions. In our case, fitting to a bi-exponential would not be meaningful due to the noise present in the off-resonance cases, which results in a large error in the extracted lifetime values. However, our approach follows all other works on single G-centers, which model the lifetime measurements with a mono-exponential function. The fact that this is correct can be seen by solving the rate equations for the carrier occupation densities at the different energy levels. The G-center can be modeled as a system featuring a ground and an excited state, and a metastable state the system can non-radiatively decay to. Considering both the radiative rate γ_R and the non-radiative rate γ_{NR} in our rate equations, and solving for the population N_2 of the excited state, we find that [Redjem, W. *Doctoral dissertation, Université Montpellier* (2019)]

$$N_2(t) \propto e^{-\frac{t}{\tau}}, \quad (1)$$

with the lifetime τ being

$$\tau = \frac{1}{\gamma_R + \gamma_{NR}}. \quad (2)$$

This means that the evolution of the excited state is modeled as a mono-exponential function even in the presence of both radiative and non-radiative decay paths. This reasoning is followed by all our cited works on G-centers. Some works (e.g. [Baron, Y. et al. *Applied Physics Letters* **121** (2022)], [Prabhu, M. et al. *Nature Communications* **14**, 2380 (2023)]) report a bi-exponential fit as well, but they show that the long decay time comes from the sample photoluminescence background, or that the time constant of the initial peak matches with that of the excitation laser.

As the Reviewer points out, it is true that in some cases the presence of both radiative and non-radiative decay channels leads to a bi-exponential modeling of the excited state evolution. This is for example the case shown in [Lin, S. D. et al. *Optics Express* **20**, 19850-19858 (2012)], where the dark state in a quantum dot can populate the excited state, effectively leading to more than one radiative decay channel. As this is not the case in our system, the mono-exponential fit is best suited to describe the excited state evolution.

.....

Reviewer 1: Moreover, have they measured their setup instrument response function and used any deconvolution algorithms to extract a more accurate lifetime fit?

Doing so could potentially illustrate the change in the decay time of the radiative transition.

Our reply: The Reviewer is correct in observing that taking into account the instrument response function might be beneficial. We indeed measured the laser and SNSPDs' response function, and convolved it with the fit function. However, the difference between the lifetime values extracted from the IRF-corrected and non-IRF-corrected fits is minimal, making the correction not significant for our conclusions. For the sake of completeness, we now attach the fits and lifetime values both here and in our manuscript (see SI, Sec. 4) and comment as follows:

IRF-corrected lifetime measurements. a-e) Lifetime data and IRF-corrected fits. f) Laser and SNSPDs' response function.

“We note that the lifetime data presented in Fig. S6 are not corrected for the laser and SNSPDs' response function. In general, it is important to take the Instrument Response Function (IRF) into account, as it may distort the signal and thus affect the reliability of the fits. For this reason, we measured the IRF and convolved it with the fit function in Eq. 5. In this way, we properly include the laser and SNSPDs' response when extracting our lifetime values. The IRF-corrected data are displayed in Fig. S7, together with the IRF. The extracted lifetime values do not vary significantly in comparison to the previous case where no IRF correction was applied.”

Reviewer 1: As I have mentioned earlier, I think this research is a big step forward in the world of quantum technology using silicon-based artificial atoms. I'm looking forward to seeing the updated manuscript and how my questions have been addressed.

Our reply: We thank once again the Reviewer for all the feedback and constructive comments, which have helped us improve the quality and clarity of our work. We hope that our manuscript can now be found suitable for publication in *Nature Communications*.

In response to Reviewer 2:

Reviewer 2: The authors have addressed many (but not all) of my criticisms. In particular, they have reduced the claims and put the work into the right perspective. However, with this reduction – albeit appropriate – I’m afraid that Nature Communications is not a suited journal, as it seems that the work is not relevant for the majority of the community. To summarize, it does not add anything to the literature concerning emitters in the solid. The findings even fail to be of relevance for the subfield working on emitters in silicon... not even for the groups working in color centers. Instead, it seems that the results are only of interest for the few groups that work on G-centers, or - even more narrow - on a sub-class of this specific defect. This narrow scope is in contradiction with the targeted scope of the journal, such that I can only recommend to resubmit the work to another journal that is more focused.

Our reply: We are thankful to the Reviewer for all their previous comments, which have helped us improve our manuscript. While we respect this final viewpoint, we believe that the impact and purpose of our work aligns with the journal’s scope and standards, as suggested by the other Reviewers as well.

Artificial atoms in silicon are ideal candidates for quantum networks, due to their emission wavelength and great prospects for scalability. However, this field is still very young, and significant effort is being put by several research groups worldwide into finding the most suitable defect in silicon. Our work adds new information to the literature, providing novel insights on the properties of G-centers when integrated into photonic cavities. Moreover, we discuss the hypothesis of two different types of artificial atoms labeled as G-centers. This hypothesis has spurred further research and discussions within the scientific community. Remarkably, the recent work [Durand, A. et al. Genuine and faux single G centers in carbon-implanted silicon. *arXiv:2402.07705* (2024)] provides evidence supporting our hypothesis.

For these reasons, we are confident that our work deserves publication, as it not only contributes to but also advances the existing literature on the state-of-the-art of silicon color centers.

In response to Reviewer 3:

Reviewer 3: The authors have carefully addressed most of my comments, and I appreciate the care and seriousness put into the response. I also find the manuscript greatly improved. However, I maintain some criticism on two points. The first may be conceptually important but perhaps not so significant for the results. The second point is more important because the claims of the paper do not match the results. The paper contains important high-quality data that clearly deserve publication, but I cannot see that Nature Communications is a proper outlet unless the authors significantly revise their claims to be consistent with their experimental findings: In contrast to their claims, their data shows that G centers are not very promising for quantum technologies and that extreme Purcell factors are needed to bring them into a regime of even remotely useful quantum efficiencies.

Our reply: We thank the Reviewer for providing helpful comments and constructive feedback, and we are happy to read that our response has been positively assessed. The Reviewer is bringing up two more points, which we are happy to address in what follows. We hope that, after having clarified these last concerns, our manuscript can be found suitable for publication in *Nature Communications*.

.....

Reviewer 3: The response provided by the authors on the discussion of the Purcell factor, $(1 + F_P) * \gamma_R$, is not correct. I now understand where it comes from, and this is approximately valid for many types of microcavities and macroscopic cavities where the spectrum of the LDOS is approximately a peak on a background. In this case, the background can be approximated by a uniform medium, which after normalization enters (in the Purcell factor) as an additional term of unity. But the authors consider a photonic crystal cavity where this approximation does not hold as the background LDOS is strongly suppressed by the (in-plane) photonic band gap, which reduces the background LDOS by 10-100x dependent upon position and polarization (and to a minor degree also on wavelength). The authors should address how this impacts their findings.

Our reply: We thank the Reviewer for bringing up this important point, which we are happy to clarify. Below we show that the used mathematical definition of the enhancement does not impact the correctness of our results. As the Reviewer points out, we have assumed the case where the emission rate outside of the cavity mode matches that in a homogeneous medium, which leads to the factor 1 in the enhancement $(1 + F_P)\gamma_R$ of the radiative rate γ_R , as thoroughly explained in the revised version. The Reviewer is arguing that this approximation is not valid in the case of our cavities, and that we should therefore have the enhancement defined as $(\alpha + F_P)\gamma_R$, with $\alpha < 1$. Therefore, this would lead to an incorrect derivation of our results. However, our results are still correct because the value of the derived quantum efficiency — which is the only result we extract using the discussed formula — is independent of the parameter α . This can be seen by expanding the formulas provided in SI, Sec. 7. Let us start by considering the rate enhancement as $(1 + F_P)\gamma_R$, as done in our study. We show in SI, Sec. 7 that

we obtain

$$QE < \frac{\frac{\tau_{\text{off}}}{\tau_{\text{off-th}}} - 1}{F_P F_{\text{DW}}} = \frac{\frac{\tau_{\text{off}}}{\tau_{\text{off-th}}} - 1}{\left(\frac{\Phi_{\text{on}}}{\Phi_{\text{off}}} - 1\right) F_{\text{DW}}} = 0.18 \pm 0.01.$$

We now consider the enhancement $(\alpha + F_P)\gamma_R$. Following the same steps, we obtain

$$QE < \frac{\frac{\tau_{\text{off}}}{\tau_{\text{off-th}}} - 1}{(F_P + \alpha - 1) F_{\text{DW}}} = \frac{\frac{\tau_{\text{off}}}{\tau_{\text{off-th}}} - 1}{\left(\frac{\Phi_{\text{on}}}{\Phi_{\text{off}}} - \alpha + \alpha - 1\right) F_{\text{DW}}} = 0.18 \pm 0.01,$$

which is the same expression as the previous one. This means that the factor 1 (or α) does not affect our findings. The full derivation of this result is now provided in SI, Sec. 7.

However, the Reviewer is correct in their observation that our case differs from other experiments. We now rewrite Eq. 1 considering the more general case of enhancement $(\alpha + F_P)\gamma_R$, and clarify this throughout our manuscript as follows:

“... into other modes:

$$\beta = \frac{(\alpha + F_P)\gamma_R}{(\alpha + F_P)\gamma_R + \gamma_0}, \quad (3)$$

where [...] $0 < \alpha < 1$ is a parameter depending on the specific device structure and geometry (see SI, Sec. 7).”

“By defining $F_P^{\text{leak}} \equiv \alpha$, we can rewrite the generalized Purcell factor as $(\alpha + F_P)$. The factor α depends on the specific cavity type and geometry.”

“... we find

$$QE < \frac{\frac{\tau_{\text{off}}}{\tau_{\text{off-th}}} - 1}{F_{\text{DW}}\left(\frac{\Phi_{\text{on}}}{\Phi_{\text{off}}} - 1\right)}, \quad (4)$$

which we use to estimate the QE to be bounded by...”

“We note that the bound on the QE is independent of the factor α dictating the radiative rate enhancement.”

.....

Reviewer 3: My second point regards the claims of the paper in relation to the quantum efficiency. I am fully aware that it has become customary to make inflated claims about paving the way for some kind of quantum technologies in nearly all papers dealing with quantum systems, but it does not make it meaningful. The implication of the author’s observations is that while the g2 looks impressive and useful, the decay is predominantly non-radiative, which is why they do not observe any enhancement of the measured decay rate. Since the measured lifetime is on the order of a few nanoseconds, the non-radiative decay must be much faster (as also discussed explicitly by the authors). In their discussion of erbium emitters, they mention their very long lifetime as a downside, and I agree, but it actually implies an

extremely small non-radiative decay rate. So it appears that erbium emitters must be greatly enhanced to increase the radiative rate and the color centers studied by the authors must be greatly enhanced to combat the large non-radiative rate. In other words, there is no obvious advantage of the G centers in this regard. Now, the paper is not about erbium so this is less important, but, on this background, a statement like “Our results show intensity enhancement of G-centers as well as highly pure and efficient single-photon emission, paving the way towards scalable quantum information processing” seems to be simply incorrect. Their results indicate on the contrary that G-centers are not very promising (modulo the possibility of the existence of several kinds of G centers) for quantum technologies. A similar comment applies to the claim of “cavity enhancement” of single-photon emission, because the data shows only reshaping of the far field through their nicely designed cavity, but no Purcell effect.

Our reply: We thank the Reviewer for raising this important point. Following the Reviewer’s suggestions, we have now rephrased our claims making it clearer that we do not observe any lifetime reduction, and that we hypothesize a strong non-radiative decay. This makes us derive novel insights into this class of emitters, gaining new information about the potential existence of two different types of G-centers. We now discuss this both in the abstract and main text. In more detail:

- The claim “*Our results show intensity enhancement of G-centers as well as highly pure and efficient single-photon emission, paving the way towards scalable quantum information processing.*” has now become

“Our results show enhancement of the G-centers’ zero phonon line intensity as well as highly pure and efficient single-photon emission, while their lifetime remains unchanged within the error. Our work suggests the possibility of two different types of G-centers treated in the literature, shedding new light into the properties of these emitters.”

- We have also modified the final part of the Introduction “*Here, we report on the inverse design of high Q/V , η -optimized photonic crystal cavities, and demonstrate cavity-enhanced interaction of light with single artificial atoms at telecommunication wavelengths in silicon.*” to

“Here, we report on the integration of single artificial atoms into inverse-designed, η -optimized photonic crystal cavities. We show cavity enhancement of the zero phonon line (ZPL) intensity, while the excited state lifetime of our atoms remains substantially unmodified. Our results suggest the possibility of two different types of artificial atoms labeled as G-centers.”

This way, we make it clearer that the enhancement concerns the zero phonon line emission of our G-centers. This confirms what has been reported for ensembles of G-centers [Lefaucher, B. et al. *Applied Physics Letters* **122**, (2023)].

- Along similar lines, we also clarify the following:

“We do not observe a statistically significant lifetime modification even under a clear enhancement of the zero-phonon emission rates above 6x.”

“We show intensity enhancement of G-centers’ ZPLs coupled to silicon nanocavities, and highly pure and efficient single-photon emission.”

- We also stress more the idea that finer spatial alignment, higher quality factors and smaller mode volumes might be needed to achieve a system with highly desirable properties for quantum information processing, such as higher coherent photon emission. Moreover, we discuss the possibility of the metastable state investigation as follows:

“As discussed in the previous section, lifetime reduction could still be achieved by obtaining finer cavity-atom spatial alignment and higher Q/V , resulting in a system with predominant radiative decay, highly suitable for quantum information processing. New directions may also involve the investigation of the anticipated spin in the metastable state, which holds promise for various quantum applications such as quantum sensing and security protocol demonstrations.”

REVIEWERS' COMMENTS

Reviewer #1 (Remarks to the Author):

After the most recent round of revision, the authors have updated their claims on the photonic crystal cavity enhancement of single G-centers in silicon. They now argue that their photonic crystals enhance the zero phonon line of the emitters, and provide a more convincing discussion that their work provides more evidence of multiple states of G-centers. They have also fully answered my previous technical questions and improved the manuscript since the last round of review. I have no additional questions and I believe this manuscript now fits the high standards for publication in Nature Communications.

Reviewer #3 (Remarks to the Author):

The authors present a thorough response to my previous questions and have made appropriate revisions to the manuscript, which I find significantly improved relative to the initial submission - I hope the authors agree. The results constitute an important contribution to the rapidly evolving research on emitters in silicon, and I recommend publication in Nature Comm.